# PHLOWER leverages single-cell multimodal data to infer complex, multi-branching cell differentiation trajectories

Mingbo Cheng [1,2,9], Jitske Jansen[3,4,9], Katharina C. Reimer [3,4], Vincent P. Grande[5], James S. Nagai [1,2], Zhijian Li [6,7], Paul Kießling [3], Martin Grasshoff[1,2], Christoph Kuppe [3,10], Michael T. Schaub [2,5,10], Rafael Kramann [2,3,4,8,10] & Ivan G. Costa [1,2,10]

Computational trajectory analysis is a key computational task for inferring differentiation trees from this single-cell data. An open challenge is the prediction of complex and multi-branching trees from multimodal data. To address these challenges, we present PHLOWER (decomposition of the Hodge Laplacian for inferring trajectories from flows of cell differentiation), which leverages the harmonic component of the Hodge decomposition on simplicial complexes to infer trajectory embeddings from single-cell multimodal data. These natural representations of cell differentiation facilitate the estimation of their underlying differentiation trees. We evaluate PHLOWER through benchmarking with multi-branching differentiation trees and using kidney organoid multimodal and spatial single-cell data. These demonstrate the power of PHLOWER in both the inference of complex trees and the identification of transcription factors regulating off-target cells in kidney organoids. Thus, PHLOWER enables inference of complex branching trajectories and prediction of transcriptional regulators by leveraging multimodal data.

Cellular differentiation, the process by which a cell changes its chromatin and expression programs to acquire more specialized functions, is not only crucial in the development of multicellular organisms but also key during onset and progression of diseases. Transcription factors (TFs), which are proteins binding to regulatory DNA regions (open chromatin regions), are key regulators of gene expression, thereby orchestrating cellular differentiation processes. Dissecting these key regulatory events will help to develop protocols for cellular reprogramming or ex vivo differentiation in, for example, organoids and to understand disease-related differentiation processes for potential therapeutic interventions[1]. In this context, a unique resource to understand the interplay between chromatin, regulatory signals (TF binding) and expression changes during cellular differentiation[2,3], is multimodal single-cell sequencing[2], which can measure both full expression programs and genome-wide open chromatin. However, the associated experimental protocols dissociate cells, making it impossible to track how an individual cell differentiates over time experimentally.

To overcome this experimental limitation, computational trajectory analysis, which explores nonlinear embeddings in the cellular space and algorithms to find trees in these spaces, has therefore become an important tool in the analysis of cell differentiation in single cells[4–7]. However, these approaches have been hitherto applied to the study of differentiation trees with small trees (three to nine branches) and the only comprehensive benchmarking is based on small trees with four to five differentiation branches[8]. Currently, no work has evaluated their scalability to complex and multi-branching trajectories. Moreover, the analysis of single-cell multiomics data provides further challenges, as there is a need to estimate joint embeddings

across modalities for modality detection[9,10]. Altogether, there is a lack of computational approaches to infer complex, branching trajectories from multimodal sequencing data.

We propose here PHLOWER. Simplicial complexes (SCs) are generalizations of graphs that allow not only for nodes and edges, but also triangles and other higher-order structures to be present[11]. The discrete Hodge Laplacian (HL) on SCs represents a generalization of the well-known graph Laplacian explored in diffusion maps[12] and trajectory inference for single-cell data[13,14]. PHLOWER uses SCs as representation of single-cell multimodal data, where a node represents a cell and an edge indicates a potential cell differentiation event. The harmonic component obtained via spectral decomposition of the HL on an SC allows the creation of edge-flow embeddings (cell differentiation events) and trajectory embeddings (cell differentiation paths)[11]. These represent cell differentiation processes directly, and can thus enable the detection of complex differentiation trees and characterize branching events with high precision.

## Results

### Differentiation tree inference with PHLOWER

PHLOWER uses the discrete HL and its associated Hodge decomposition to obtain embeddings of cell differentiation trajectories. The Graph Laplacian (a zero-order HL) is a matrix representation of graphs, where samples are encoded as vertices and distances as edge weights. Particular eigenvectors of the Graph Laplacian are used to obtain an embedding representing cells in the graph, forming the basis for methods in clustering analysis[15], nonlinear dimension reduction[12], trajectory inference and pseudotime estimation in single-cell data[13,14]. In PHLOWER, we represent single-cell data as an SC, that is, a higher-order generalization of a graph, consisting of nodes (0-simplices), edges (1-simplices) and triangles (2-simplices). The spectral decomposition of the HL can be used to decompose edge flows into gradient-free, curl-free and harmonic components[11,16]. Of note, while the HL and Hodge decomposition are defined on differential forms on Riemannian manifolds[17], there are guarantees that the behavior of the HL on an SC converges to the HL on manifolds in the limit[18,19]. PHLOWER focuses on the harmonic eigenvectors of the HL[11,16] as these are associated with holes in the SC, which in turn can reveal cell differentiation tree branches in the gene expression SC.

In short, PHLOWER first uses a graph representation of the single-cell data and the graph Laplacian to estimate pseudotime of cells and identify progenitor and terminal cells, similarly to methods in previous studies[13,20]. Cell differentiation processes are classically represented as generating an out-branching, tree-like structure. To transform such a directed branching process into a SC, we perform a Delaunay triangulation and connect terminal differentiated cells (high pseudotime) with progenitor cells (low pseudotime). Connecting cells with high and low pseudotime creates a hole for every main trajectory in the graph (Fig. 1). Next, we perform a Hodge decomposition of the edge-flow space on the SC. The harmonic components of this decomposition provide edge-level embeddings, where each point in an embedding represents a cell differentiation event (equation (19)), as well as a trajectory embedding, where each embedding point represents a cell differentiation trajectory (equation (21)). PHLOWER explores these embeddings to delineate major differentiation trajectories in single-cell data and to reconstruct complex cell differentiation trees. As an example, we show how PHLOWER can be used to predict a simple differentiation tree from mouse embryonic fibroblasts (MEFs)[21] toward either neurons or myocytes (Fig. 1 and Extended Data Fig. 1).

**Benchmarking of trajectory inference methods.** Next, we evaluated PHLOWER and competing methods on simulated datasets[22] and single-cell RNA-sequencing (scRNA-seq) datasets[8] on how well methods can recover original tree structures and how well they can place cells within these differentiation trees. We utilized the diffusion-limited

aggregation (DLA) tree[22] to simulate ten complex differentiation tree datasets with 5 to 18 total branches, similarly to previous work[14]. We selected 33 scRNA-seq datasets from the benchmarking Dynverse, which contained single-rooted tree structures with at least three branches (Supplementary Table 1).

These data were provided as input for PHLOWER and competing approaches for the detection of differentiation trees (PAGA tree[4], Monocle3 (ref. 23), cellTree[24], pCreode[25], Slice[26], RaceID[27,28], Slingshot[6], TSCAN[29], MST[30], Elpigraph[31] and STREAM[5]). This selection included the top ten approaches from a recent benchmark evaluation[8]. Tools were evaluated using the metrics proposed by the Dynverse framework[8]: tree structure similarity (Hamming–Ipsen–Mikhailov, HIM), location of cells within a branch (correlation), cell allocation to branches (F1 branches), cell allocation to branching points (F1 milestones) and an accuracy estimated as the average of the four previous metrics. An example of the steps of the PHLOWER algorithm in fitting a differentiation tree in simulated data is shown in Supplementary Fig. 1.

For simulated data, PHLOWER was the best-performing method in regard to tree topology recovery, followed by PAGA, RaceID and Monocle3 (Fig. 2a and Extended Data Fig. 2a). Regarding the problem of allocating cells to positions in a branch, PHLOWER was also the best performer, followed by TSCAN and RaceID. In allocating cells to the correct branches, PHLOWER obtained the best performance, followed by Monocle3 and PAGA. In the allocation of cells to branches, PHLOWER was the best performer, followed by RaceID, Monocle3 and PAGA. The final accuracy value indicated PHLOWER, PAGA and RaceID as the best-performing methods for the simulated data.

For real scRNA-seq data, PHLOWER was the best-performing approach followed by Monocle3 and PAGA regarding structure similarity (Fig. 2b and Extended Data Fig. 2a). Regarding the location of cells, PHLOWER obtained highest average scores followed by Monocle3 and Slingshot. Regarding allocation to branches or millstones, PHLOWER was also the top performer with Slice, and Slingshot as follow-up approaches. In the aggregated accuracy ranking, PHLOWER obtained the highest average value followed by Monocle3 and pCreode.

The ranking of competing approaches differed on simulated versus scRNA-seq datasets. While RaceID, Monocle3 and PAGA were runners-up on simulated data, Monocle3, pCreode and Slingshot were runners-up for real data. One potential reason is that these methods perform differently in distinct types of tree structures, that is, simulated data focus on larger complex tree structures. To further investigate this, we stratified the performance statistics on real scRNA-seq by type of structure as determined by Dynverse (bifurcation, multifurcation and trees)[8]. This indicates that some approaches, such as Slingshot, perform relatively better in simpler trees (bifurcation), while others perform relatively better on complex structures (monocle and PAGA; Extended Data Fig. 2b). These results are also supported by an analysis showing that some approaches, such as Slingshot, tend to underestimate the size of the trees (Extended Data Fig. 2c).

Another important aspect is the computational requirements of the approaches. To evaluate this, we resorted to two datasets commonly used in the RNA velocity literature: the pancreas progenitor (≈3,700 cells)[32] and neurogenesis (≈18,000 cells) datasets[33]. PHLOWER required 0.5–12 h and 12–40 GB of memory for these datasets using a standard desktop (Extended Data Fig. 3a,b). We next evaluated a time-efficient variant of PHLOWER based on cell downsampling (Extended Data Fig. 3c,d). We observed that by using 30% of the cells, PHLOWER obtained 8.6 times faster processing and used one-sixth of the memory in contrast to using all cells, while infering a bona fide neurogenesis cell differentiation tree.

The performance of PHLOWER and competing methods is further illustrated by the analysis of the inferred trees in the pancreas progenitor and neurogenesis datasets. We focus here on PHLOWER, PAGA and Monocle3, as these were the only methods recovering complex trees, when compared to other approaches (Supplementary Figs. 2 and 3).

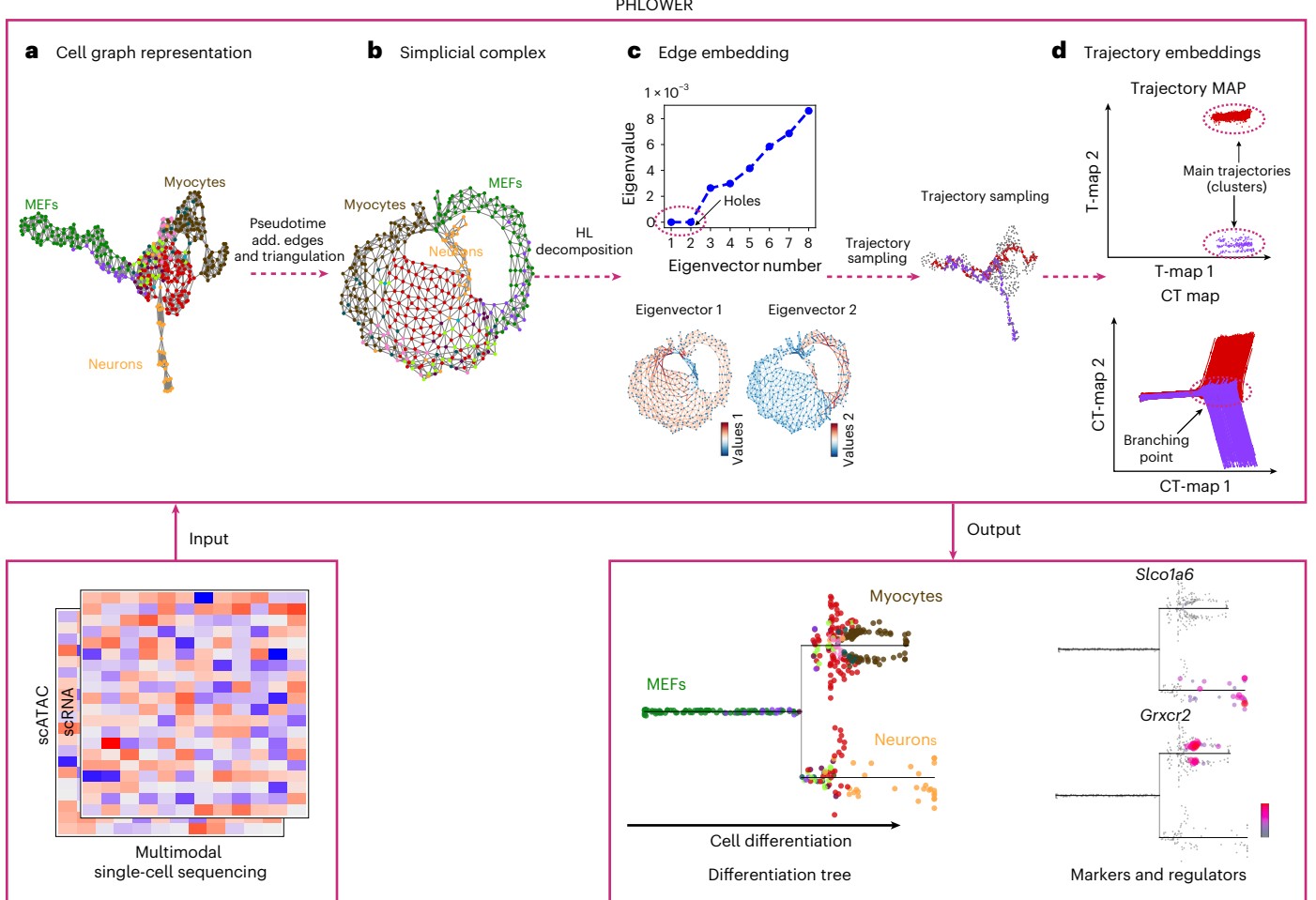

**Fig. 1 | Schematic overview of major steps of PHLOWER.** PHLOWER receives as input a multimodal or a unimodal single-cell dataset. **a**, It next creates a graph representation and a graph embedding[62] of the cells. Our example is based on a differentiation system of MEFs toward neurons or myocytes[21]. From this, PHLOWER uses a graph Laplacian decomposition and a random walk to estimate pseudotime from progenitor cells (that is, MEFs). **b**, Next, cells with low pseudotime (progenitors) and high pseudotime (terminally differentiated cells) are connected and a simplicial complex is obtained by Delaunay triangulation of the graph. **c**, An edge embedding is obtained via the harmonic eigenvectors via the decomposition of the simplicial complex HL. In the MEF data, the

first two eigenvectors have zero eigenvalues and their signals discriminate edges belonging to the two differentiation trajectories (or holes in the SC). **d**, PHLOWER performs the next random walks to obtain trajectories and uses the HL decomposition to obtain a trajectory embedding and a cumulative trajectory (CT) embedding. Clustering analysis in the trajectory embedding reveals major trajectories in the data, while the cumulative embedding space is used to estimate trajectory backbones and branching-point events. PHLOWER outputs stream trees representing the differentiation tree over pseudotime and allows the detection of regulators and gene markers.

This is in line with the previous analysis, which indicates that most competing approaches underperform in complex differentiation trees (Extended data Fig. 2c).

For pancreas progenitor data (Fig. 2c), we observed that the Monocle3 tree does not capture the epsilon branch, and the delta branch corresponds to an unconnected single-branched tree. PHLOWER and PAGA recovered all main branches of this tree: epsilon, delta, alpha and beta. In the larger and more complex neurogenesis data (Fig. 2d), Monocle3 inferred four unconnected trees delineating some of the main branching events. PHLOWER and PAGA recapitulated the main terminal branches with the exception of the small oligodendrocyte precursor cell population, which was missed by both methods. Note however that PAGA inferred two false positive branches (Fig. 2d), which are not related to any cell type described in this dataset[34]. Altogether, this analysis supports the power of PHLOWER in recovery on complex cell differentiation trees.

An alternative approach to characterize topological features in a dataset is persistent homology (PH)[35]. To check its association with PHLOWER, we performed PH in the the triangulated

SC (Extended Data Fig. 4). This indicates PH can support PHLOWER by determining thresholds to build the triangulated SC. Moreover, a PH analysis of the SC used by PHLOWER indicated that PH can be used to characterize the number of holes in PHLOWER's SC (Extended Data Fig. 5).

**Inferring cellular trajectories in kidney organoids.** Induced pluripotent stem (iPS) cell-derived kidney organoids represent a solid model to validate cellular trajectories because this model represents the differentiation of stem and progenitor cells toward various kidney cell lineages. Therefore, we generated a multimodal single-cell multiome sequencing (RNA and assay for transposase accessible chromatin (ATAC)) datasets of kidney organoids after 7, 12, 19 and 25 days of differentiation by using our own protocol[36]. This recovered 13,751 cells with an average of 10,378 RNA transcripts and 19,263 DNA fragments (ATAC) per cell (Extended Data Fig. 6a). We next integrated the data for each modality independently[37] and used MOJITOO to obtain a joint ATAC–RNA embedding[9]. We provided the data as input for PHLOWER, which recovered a trajectory with nine branches (Fig. 3a

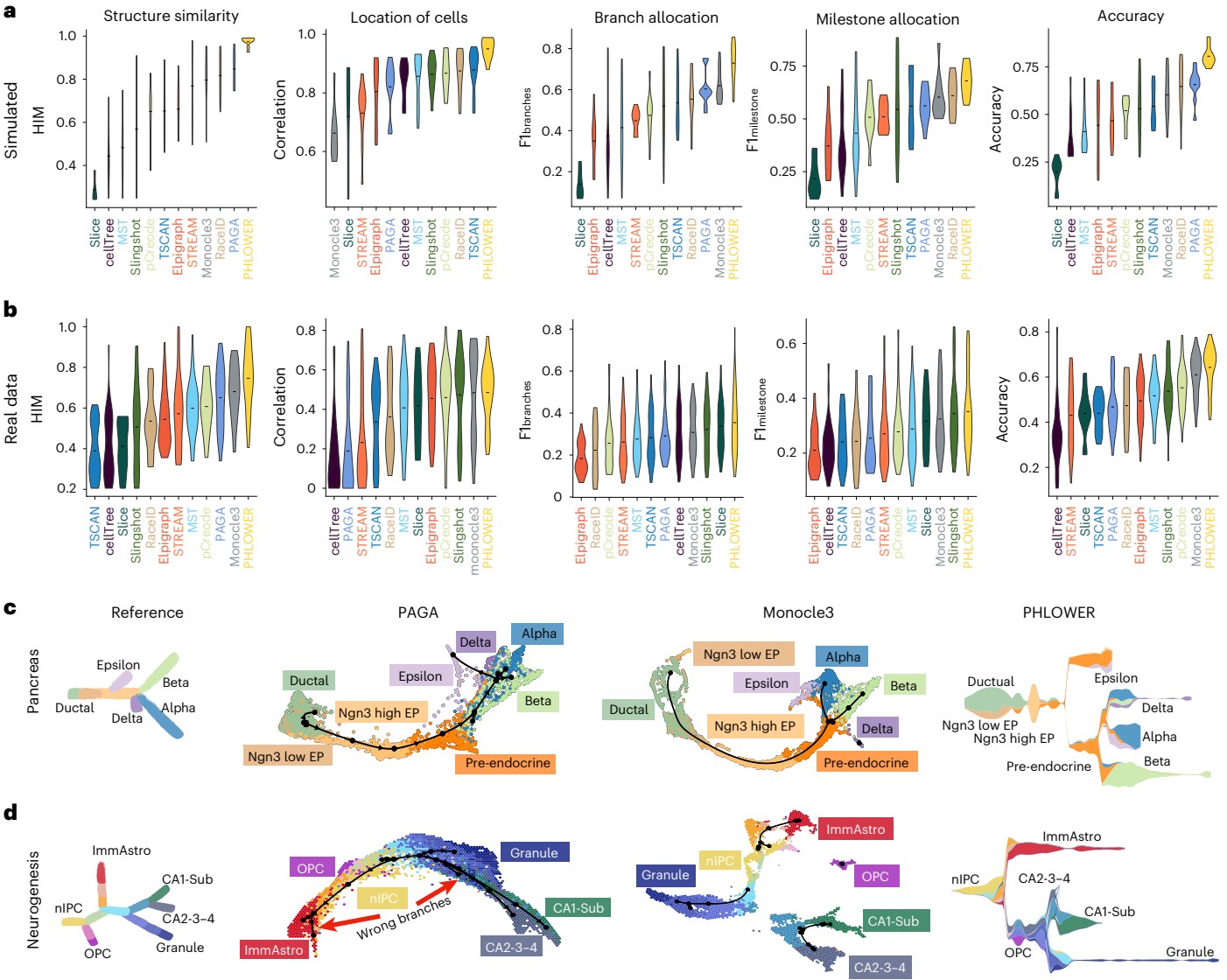

**Fig. 2 | PHLOWER benchmarking. a**, HIM score (topology similarity between true and inferred trees), cell allocation within a branch (correlation), cell allocation to branching points (F1 milestones), cell allocation to branches (F1 branches) and accuracy (y axis) after evaluation of 12 tree inference algorithms (x axis) on the simulated data (n = 10). Methods are ranked (left to right) regarding the mean value (black trace). **b**, As in **a** on Dynverse real datasets (n = 33). **c,d**, Differentiation trees estimated with PAGA, Monocle3 and PHLOWER algorithms for the pancreas progenitor (n = 3,696) (**c**) and neurogenesis (n = 18,213) (**d**) datasets. Trees in the left display the reference cell differentiation tree. Red arrows indicate false positive branches.

and Extended Data Fig. 6). We observed that the tree successfully sorted cells based on the organoid's age (Fig. 3b), and this matched well with the pseudotime estimates (Fig. 3c). These nine branches could be grouped into three major branches associated with epithelial cells (two sub-branches associated with podocytes and one with tubular cells), four branches of stromal cells and one major branch associated with muscle and neuronal cells (three sub-branches). The annotation of these cells was based on the expression of markers as *TBXT*[38] and *KDR*[39] for mesoderm cells, *PODXL*[40] and *NPHS2* (ref. 40) for podocytes, *SLC12A1* (ref. 41) and *PAPPA2* (ref. 41) for kidney tubule epithelial cells, *COL1A2* (ref. 40) and *PDGFRB*[42] for stromal cells, and neuronal and muscle markers *MAP2* (ref. 36), *MSX1* (refs. 38,40–42), *MYL1* and *MYF6* (Extended Data Fig. 7). The latter branches, that is, neuronal and muscle, are considered off-target cells and potentially may hamper maturation of kidney cells in the organoids, which should be avoided in kidney organoids[43,44].

A key question to be addressed by PHLOWER is the detection of regulators (TFs) driving the differentiation of iPS cells within the kidney

organoids. We leveraged the tree inferred by PHLOWER and a procedure similar to scMega[45] to detect TFs related to branch differentiation. In short, we estimated TF activity scores and selected TFs whose expression levels are concordant with the TF activity and are differentially expressed between compared branches (Extended Data Fig. 8). When comparing cells at the end of tubular and podocyte trajectories, this recovered bona fide regulators of these cells such as *WT1* (ref. 46) and *MAFB*[47] for podocytes and *HNF1B*[48] and *GRHL2* (ref. 49) for tubular cells (Fig. 3e and Extended Data Fig. 9). Next, we compared major branches: stromal cells versus others and neuronal/muscle cells versus others. We detected TFs related to fibroblasts including *TWIST1* and *RUNX2* (refs. 50,51) as regulators of stromal cells, and the known skeletal muscle TF *MYOG* for the muscle branch. Among the top three neuronal cell regulators, we observed the TFs *PAX3*, *RFX4* and *ZIC2*, which have been previously related to neuronal cell differentiation and/or neuronal diseases[52–54]. The modulation of these TFs is of particular interest, as neuronal cells are considered to be undesired off-target cell types in kidney organoids.

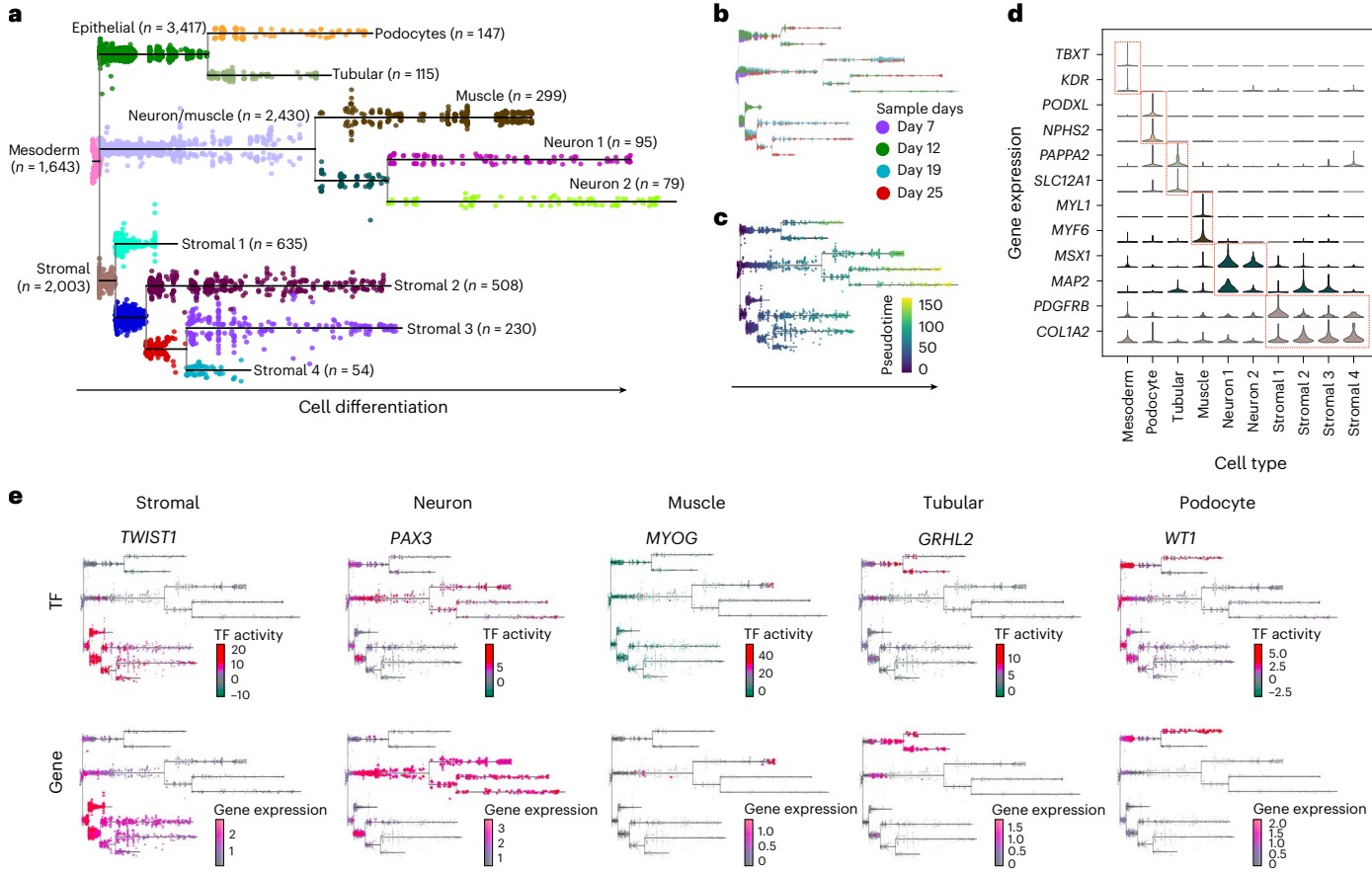

**Fig. 3 | Kidney organoid data. a**, Differentiation tree on the kidney organoid data as estimated by PHLOWER ($n = 13,751$). **b,c**, Day of differentiation (**b**) and pseudotime estimates (**c**) of cells in the differentiation trees ($n = 13,751$). **d**, Violin plots show the marker genes for cell types in each branch. **e**, Selected TFs with branch-specific expression and TF activity scores ($n = 13,751$).

**Spatial organization of kidney organoids and perturbation of off-target lineages.** We leveraged our analysis of multimodal single-cell data to perform a subcellular resolution spatial transcriptomic analysis of kidney organoids using the Xenium spatial imaging platform[55]. We used PHLOWER results to derive a 100-gene kidney organoid marker panel for the detection of mesenchymal cells, tubular epithelial cells, stromal cells, neuronal cells and podocytes (Extended Data Fig. 10a). This panel selection included computational data-driven-based marker genes from the multiome analysis, some literature-based marker genes and top candidate TFs from the scMega analysis (Supplementary Table 2).

Clustering and trajectory analyses of the 105,092 cells in a 19-day kidney organoid and two 25-day kidney organoids indicated the identification of progenitor cells (mesoderm, podocyte and tubular epithelial progenitors) and all major kidney organoid branches (mesoderm, podocytes, tubular epithelial cells, stromal cells and neuronal cells; Fig. 4a,b and Extended Data Fig. 10b,c). When contrasting cells detected on 19 versus 25 days, we observed a higher proportion of mesoderm cells on day 19, while day 25 had a higher amount of tubular epithelial cells and podocytes (Fig. 4c), reflecting the higher maturation of organoids on day 25. Moreover, we observed gene and TF expression patterns similar to the predictions based on the multimodal analysis (Fig. 4d, Extended Data Fig. 10d and Supplementary Figs. 4 and 5). This included the TFs *PAX3*, *RFX4* and *ZIC2*, which were predicted by PHLOWER to control the off-target neuronal lineage. Altogether, these results indicate that the Xenium panel can identify major differentiation branches in kidney organoids in a spatial context and will aid in understanding cell differentiation differences based on signaling events originating from cellular neighborhoods and spatial niches.

To validate PHLOWER predictions, we performed multiplex short interfering RNA (siRNA) knockdown experiments of the identified neuronal lineage-defining TFs *PAX3*, *RFX4* and *ZIC2* in iPS cell-derived kidney organoids during the differentiation process from day 19 onwards. Knockdown experiments led to a reduction of 25–30% in *PAX3*, *RFX4* and *ZIC2* mRNA expression (Extended Data Fig. 10e). Xenium spatial profiling of scrambled siRNA as the control condition (scrambled; sections 1 and 2) and siRNA-treated organoids against *PAX3*, *RFX4* and *ZIC2* detected the same cell types as non-treated day-25 organoids (Fig. 5a and Supplementary Figs. 6 and 7). When contrasting scrambled siRNA versus siRNA, we observed a significant increase in tubular cells and podocyte progenitors and a significant decrease in muscle, stromal and neuronal cells (Fig. 5b). Immunofluorescence staining further confirmed the reduction of stromal cells including off-target neuronal cells in the siRNA condition, as reflected by the significant decrease in interstitial vimentin (Fig. 5c,d). We observed a significant increase in podocyte nephrin protein expression and 50% more tubular E-cadherin protein expression, suggesting improved podocyte (progenitor) and tubular characteristics as a result of diminished off-target cells (Fig. 5e,f). Altogether, these experiments supported that gene silencing of *PAX3*, *RFX4* and *ZIC2*—known as neuronal lineage markers and considered as an off-target population—led to a small increase in tubular cell proportion and enhanced podocyte development as shown by increased nephrin and a higher proportion of podocyte progenitors.

## Discussion

Trajectory analysis is paramount in the analysis of cells undergoing cell differentiation. Despite a wealth of literature on computational methods[8], most methods have been only applied or tested in small cell

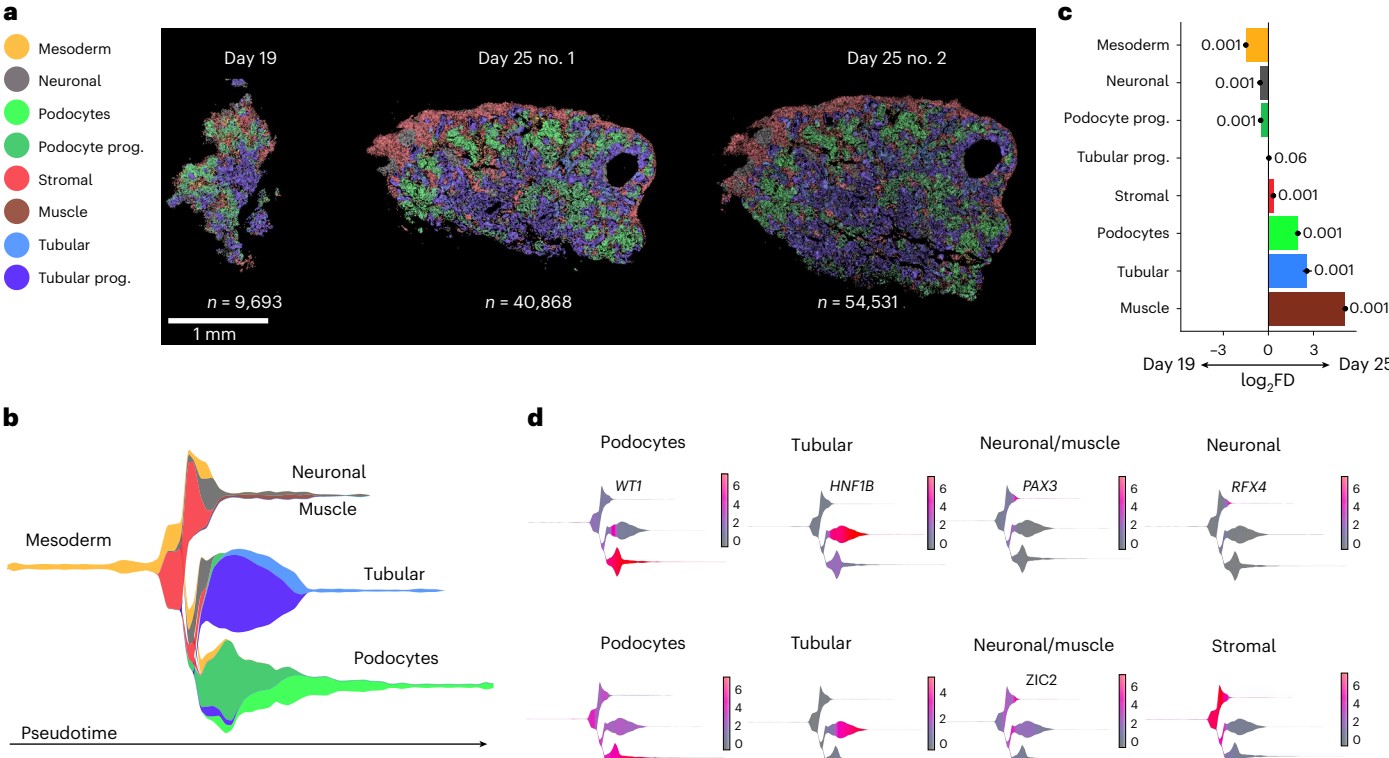

**Fig. 4 | Spatial profiling of kidney organoids. a**, Xenium-based spatial profiling of kidney organoid differentiation on day 19 and day 25 (sections 1 and 2). Colors represent distinct cell populations. **b**, PHLOWER-estimated differentiation tree on day-19 and day-25 organoids ($n$ = 105,092). **c**, Single-cell permutation test (two sided) to measure relative differences in cell proportions comparing cell abundances on day 25 versus day 19. $x$ axis shows the $\log_2$-transformed fold difference ($\log_2$FD), error bars are based on confidence intervals (95%) for the magnitude difference returned via bootstrapping[63] ($n$ = 1,000), and $P$ values were adjusted using a Benjamini–Hochberg false discovery rate[64]. **d**, Expression of TFs controlling the differentiation of the kidney organoid branches.

differentiation trees with few branches. PHLOWER explores the harmonic component of the HL decomposition, which allows the consideration of the interaction between edges, nodes and triangles of an SC to estimate embedding on edges and trajectory levels. By exploring this higher-order representation of the data explicitly, it allows the detection of complex and large cell differentiation trees. This is supported by our comprehensive benchmarking, which indicates that PHLOWER obtains highest average scores in all evaluated scenarios and metrics. Of note, our evaluation focused on single-rooted tree structures with 3 to 26 branches. This is in contrast to a previous benchmarking study[8], which includes linear or circular structures.

SCs represent higher-order generalizations of graphs. Therefore, computations on SCs can be computationally demanding owing to combinatorial explosion. While the SCs explored by PHLOWER are sparse, which results in relatively few edges and triangles, PHLOWER required downsampling strategies to handle large spatial transcriptomics datasets with more than 100,000 cells. This highlights a current gap in efficient methods for working with SCs on large-scale data.

Multimodal single-cell data, which allows the parallel measurement of transcriptome and open chromatin data, can provide rich information on the relationship between the regulatory function and the transcriptional function of cells[3]. We display the power of PHLOWER by analyzing a kidney organoid differentiation course, where PHLOWER was able to detect major cell lineages, as well as TFs with branch-specific gene expression and TF activity. Off-target cell populations compromise organoid development and maturation[43,44,56]. To address this, we targeted TFs identified by PHLOWER as regulators of neuronal lineages using siRNA. This pertubation promoted enhanced tubular cell differentiation. Altogether, these findings highlight PHLOWER's capacity

to reveal key regulatory factors within complex cell differentiation systems and its utility for improving organoid protocols.

An approach to characterize topological features in a dataset is PH[35]. An evaluation of PH analysis of PHLOWER-generated SCs indicates that PH supports the analysis of the cell differentiation trees by determining thresholds for building the triangulated SC or to determine the number of holes. However, distinct from the harmonic components of the HL, PH does not give unique generators that quantify the relation of all simplices (edges and nodes) to the holes[57]. The combination of these approaches represents an interesting topic of further research.

HLs have been previously explored in other molecular problems[58]. HL of vector fields have been performed in RNA velocity estimates, which allowed visualization of curl-free, divergent-free and harmonic components[59]. This work also suggests that the harmonic component captured the overall direction of differentiation similarly as explored in PHLOWER. However, the previous work focused only on the visualization of the RNA velocity fields, and could not be used to infer the trees or to allocate cells along these trees as performed by PHLOWER. A current limitation of PHLOWER is that it only considers the harmonic components of the Hodge decomposition. Incorporating additional components, such as curl and gradient terms, would enable the analysis of more complex cell differentiation structures, including acyclic graphs. Future applications may also involve three-dimensional molecular data, such as protein–ligand prediction[60] or three-dimensional spatial transcriptomics[61]. In such geometric data, higher-order structures, such as the tetrahedron (3-simplices), can be analyzed, which may reveal geometric features (2-loops) like hulls or voids. Addressing these cases will require efficient numerical methods for higher-order HL decompositions.

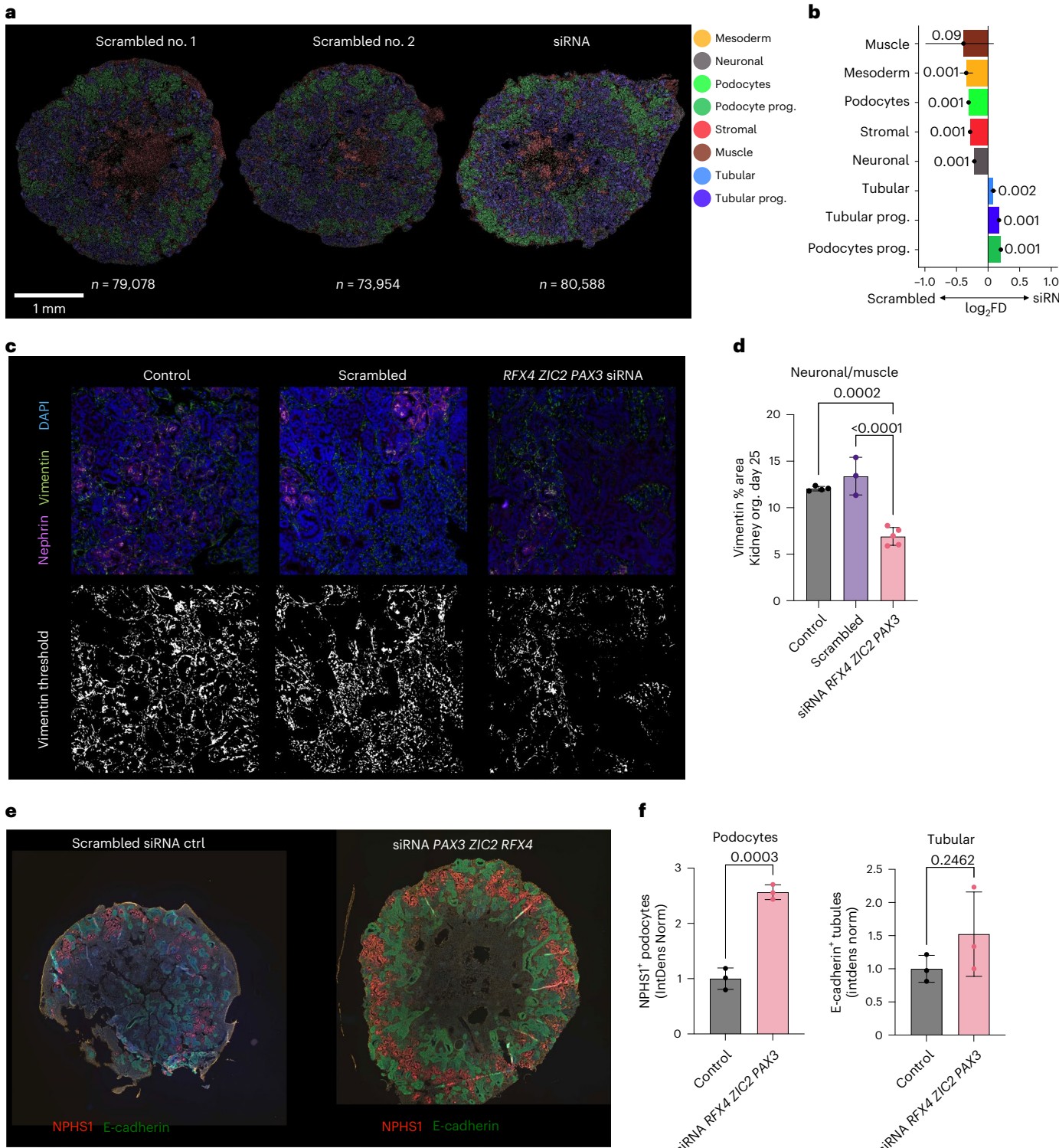

**Fig. 5 | Spatial profiling of kidney organoids after siRNA knockdown of *PAX3*, *RFX4* and *ZIC2*. a**, Xenium-based spatial profiling of kidney organoids at day 25 after treatment with scrambled siRNA control (scrambled) and siRNA multiplex against the TFs *PAX3*, *RFX4* and *ZIC2* (siRNA). **b**, Single-cell permutation test (two sided) to measure relative differences in cell proportions comparing cell abundances in siRNA and scrambled siRNA kidney organoids. *x* axis shows the log$_2$FD, and error bars are based on confidence intervals (95%) for the magnitude difference returned via bootstrapping[63] (*n* = 1,000), and *P* values were adjusted using a Benjamini−Hochberg false discovery rate[64]. **c**, Immunofluorescence staining of control, scrambled siRNA and siRNA-treated kidney organoids

showing nephrin (magenta), vimentin (green) and DAPI (blue). **d**, Quantification of the levels of neuronal marker vimentin$^+$ and NPHS1$^-$ of images in **c** using one-way analysis of variance followed by Tukey's post test with bars representing the mean ± s.d. of at least *N* = 3 organoids per condition per experiment, from two independent experiments. **e**, Silencing of *RFX4*, *ZIX3* and *PAX3* increased nephrin (red) expression in podocytes and increased E-cadherin (green) expression in tubular cells by 50%, although this was not significant. **f**, Podocyte and tubule quantification using an unpaired, two-tailed *t*-test with bars representing the mean ± s.d. of at least *N* = 3 per condition per experiment, from two independent experiments.

## Online content

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

[1]Institute for Computational Genomics, RWTH Aachen University, Aachen, Germany. [2]Center for Computational Life Sciences, RWTH Aachen University, Aachen, Germany. [3]Institute for Experimental Internal Medicine and Systems Biology, RWTH Aachen University, Aachen, Germany. [4]Department of Medicine 2 (Nephrology, Rheumatology, Clinical Immunology and Hypertension), RWTH Aachen University Medical Faculty, Aachen, Germany. [5]Department of Computer Science, RWTH Aachen University, Aachen, Germany. [6]Gene Regulation Observatory, The Broad Institute of Harvard and MIT, Cambridge, MA, USA. [7]Molecular Pathology Unit, Krantz Family Center for Cancer Research, Massachusetts General Hospital, Boston, MA, USA. [8]Department of Internal Medicine, Nephrology and Transplantation, Erasmus Medical Center, Rotterdam, the Netherlands. [9]These authors contributed equally: Mingbo Cheng, Jitske Jansen. [10]These authors jointly supervised this work: Christoph Kuppe, Michael T. Schaub, Rafael Kramann, Ivan G. Costa. ✉e-mail: rkramann@ukaachen.de; ivan.costa@rwth-aachen.de

## Methods

### Rationale

PHLOWER uses the HL and the harmonic component of the associated Hodge decomposition to obtain embeddings of cell differentiation trajectories. For this, PHLOWER represents single-cell data as an SC consisting of nodes (0-simplices), edges (1-simplices) and triangles (2-simplices). Next, a first-order HL is used to describe the SC. The first-order HL describes how edges relate to each other via nodes (so-called lower-adjacency) and triangular faces (upper-adjacency). Importantly, the decomposition of the HL can be used to decompose flows and trajectories on the edges into gradient-free, curl-free and harmonic components, akin to the classical Helmholtz decomposition of vector fields known from vector calculus. While we focus here on 'discrete' HLs based on SCs, there are guarantees that the spectral behavior of these converges to the HL on manifolds in the limit if weighted accordingly[18,19].

Of particular interest in our context here are the spectral embeddings associated with the so-called harmonic eigenvectors of the HL[11,16], as these are associated with 'holes' in the underlying space. Just like the eigenvectors of the graph Laplacian can be used to define a spectral embedding of the nodes in a graph, the eigenvectors of the HL can be used to provide an embedding of the edges. As edges correspond to cell differentiation events, this enables PHLOWER to represent cell differentiation events and cell differentiation pathways (sequences of edges) in a direct way. See Supplementary Fig. 8 for a graphical contrast of the L0 and L1 Laplacian decompositions.

### Overview of PHLOWER

PHLOWER receives as input a set of matrices representing a single-cell sequencing modality, as shown in equation (1):

$$\mathcal{X} = \{X^{(1)}, \cdots, X^{(m)}\} \tag{1}$$

where $X^{(i)} \in \mathbb{R}^{n \times s^{(i)}}$ represents the data of a particular single-cell modality, $n$ represents the number of cells, and $s^{(i)}$ represents the number of features in modality $i$. Here, we assume that cells match across modalities.

PHLOWER has been evaluated on multimodal single-cell (ATAC and RNA) sequencing or unimodal scRNA-seq ($m = 1$). First, PHLOWER constructs a graph representation $G$ of the single-cell matrix by estimating a joint embedding[9] followed by kernel representation, graph construction and pseudotime estimation as done in the literature[20]. Next, it builds an SC. For this, it first uses the Delaunay triangulation procedure between nodes (cells) to obtain edges forming triangles. Next, it uses pseudotime information from the graph to find terminal differentiated cells and root cells. Finally, it connects terminal (differentiated) cells to root (progenitors) cells with edges and triangles to create holes corresponding to cell trajectories to obtain the final SC. Thus, PHLOWER created different cell differentiation paths measurable by topology/geometry, which we will exploit in the next steps.

PHLOWER next computes the harmonic eigenvectors of the normalized first-order HL[11] associated with the SC. Eigenvectors with zero eigenvalues (harmonic eigenvectors) delineate holes in the SC, which are associated with cell differentiation trajectories. Lastly, PHLOWER samples trajectories from the SC, which represent edge flows. Taking the dot product of individual edge flows with the harmonic eigenvectors creates a trajectory embedding, where each point represents a particular trajectory. Clustering analysis on this space allows PHLOWER to find major trajectory groups. Moreover, PHLOWER builds a cumulative trajectory space, which is used to recover the differentiation trees. This can be visualized as stream plots[5]. PHLOWER outputs a tree structure, the association of each cell within a branch and pseudotime estimates (Extended Data Fig. 1). It also detects branch-specific regulators by detection of TFs similarly to ref. 45 with (1) branch-specific expression patterns and (2) similar gene expression and TF activity along the cell differentiation.

### Single-cell graph representation

As a first step, PHLOWER estimates a single-cell graph to represent the data using a procedure similar to DDHodge[20]. This procedure takes as input a low-dimensional embedding $X^t$, which can be provided by MOJITOO[9] for multimodal data; or by principal component analysis (PCA) for scRNA-seq.

**Diffusion map.** Given a common joint cell embedding $X^t$, we represent the data using diffusion maps[12]. For this, we first estimate a Gaussian kernel $W$ as shown in equation (2):

$$W_{ij} = \exp\left(-\frac{\| x_i^t - x_j^t \|^2}{\sigma_i \sigma_j}\right) \tag{2}$$

where $x_i^t$ is the representation of sample (cell) $i$ in the embedding $X^t$ and $\sigma_i$ is the local scaling parameter. This is estimated with the distance to the $n$-th nearest neighbor from $x_i^t$ as in ref. 65.

The Graph Laplacian $L_0$ can be defined according to equation (3):

$$L_0 = D - W \tag{3}$$

where $D$ is a diagonal matrix with $i$-th entry $d_{ii} = \sum_j W_{ij}$. We now consider a random-walk process on the graph described by the above graph Laplacian as shown in equation (4):

$$p_{t+1} = p_t D^{-1} W \equiv p_t M \tag{4}$$

where $p_t \in \mathbb{R}^n$ is the normalized probability vector at time $t$, and $M$ is the transition matrix of the random walk. Next, we perform an eigen-decomposition on the symmetric form of $M$ according to equation (5):

$$M' = D^{\frac{1}{2}}(M)D^{-\frac{1}{2}} = D^{\frac{1}{2}}(D^{-1}W)D^{-\frac{1}{2}} = D^{-\frac{1}{2}}WD^{-\frac{1}{2}} = Q\Lambda Q^\top \tag{5}$$

Taking advantage of the above eigen-decomposition, we can effectively calculate $M^s$ (ref. 12), as given by equation (6):

$$M^s = \left(D^{-\frac{1}{2}}M'D^{\frac{1}{2}}\right)^s = \left(D^{-\frac{1}{2}}Q\Lambda Q^\top D^{\frac{1}{2}}\right)^s = D^{-\frac{1}{2}}Q\Lambda^s Q^\top D^{\frac{1}{2}} \tag{6}$$

where the columns of $D^{-\frac{1}{2}}Q$ include the right eigenvectors of $M$ and the rows of $Q^\top D^{\frac{1}{2}}$ include the left eigenvectors. The diagonal matrix $\Lambda$ contains the corresponding eigenvalues. Finally, we can estimate pseudotime $u$ (or potential) at time $t = s$ according to equation (7):

$$u = (1/m, 1/m, \cdots, 1/m, 0, 0, \cdots, 0)M^s \tag{7}$$

where $m$ is the number of start cells, and $s$ is the step of the diffusion process.

### Graph-based pseudotime estimation

To improve the pseudotime estimates, we use a procedure similar to that used in ref. 20, which smooths pseudotime estimates by considering the graph connectivity. Let us denote the fully connected graph by $G^F = (\mathcal{V}, \mathcal{E}^F)$ and the associated pruned $k$-nearest-neighbor graph by $G^P = (\mathcal{V}, \mathcal{E}^F)$. Next, we prune this fully connected graph by considering only the $k$-nearest neighbors. This provides a graph $G^{(knn)} = (\mathcal{V}, \mathcal{E})$, where $\mathcal{V}$ is the set of vertices and $\mathcal{E}$ is the set of edges. We represent the two graphs as incidence matrices $\mathbf{B_1}^F$ and $\mathbf{B_1}^P$. For $\mathcal{X} \in \{F, P\}$, vertex $v_j \in \mathcal{V}$ and edge $e_i \in \mathcal{E}^{\mathcal{X}}$, we have equation (8):

$$B_1^{\mathcal{X}}[i,j] = \begin{cases} -1 & \text{if edge } e_j \text{ leaves vertex } v_i \\ 1 & \text{if edge } e_j \text{ enters vertex } v_i \\ 0 & \text{if otherwise.} \end{cases} \tag{8}$$

We now define initial edge weights as gradient values of the pseudotime estimates of equation (7) on the fully connected graph, for example, $w_{ij}^F = u_j - u_i$. We want to estimate pseudotimes of the pruned $k$-nearest neighbor graph by using the pseudotime estimates of the full graph. For this, we get a first estimate of the gradients of the truncated graph ($w^P$) by minimizing as shown in equation (9):

$$w^P = \arg\min_{w_P} \| \mathbf{B_1}^F w^F - \mathbf{B_1}^P w_P \|^2 + \lambda \| w_P \|^2, \qquad (9)$$

where $\lambda$ is the regularization parameter. Next, we update the potential of the vertices according to equation (10):

$$u^s = \arg\min_{u'} \| (\mathbf{B_1^P})^\top u' - w^P \|^2. \qquad (10)$$

This allows us to estimate an updated gradient $w^s = (\mathbf{B_1^P})^\top u^s$.

With the graph $G^{(\text{knn})} = G^P$ with edge weights $w^s$ and pseudotime estimates $u^s$, we estimate a graph embedding with the stress majorization layout algorithm[62,66]. Examples of the graph embeddings and pseudotime estimates are available in Extended Data Fig. 1a.

### HL decomposition on a single-cell SC

Given $G^{(\text{knn})}$ and pseudotime estimate $u^s$, the next steps are the creation of an SC and estimation of the Hodge decomposition.

**Triangulation of the single-cell graph.** A typical cell differentiation graph has tree-like structures. The harmonic eigenvectors of the HL are able to characterize distinct types of topological structures (connected components, holes and voids), but not trees or branches. PHLOWER resorts to a trick, that is, it uses pseudotime estimates to find end-state cells and adds dummy/artificial edges from end-state to start-state cells. This creates holes in the data, one for each trajectory group in the tree. For this, we obtain $m$ (default 5) cells with the highest and lowest pseudotime. Next, we use a Delaunay triangulation[67] on the graph embedding to create a hole-free set of triangles. Next, we remove edges connecting distant vertices, that is, we estimate the distribution of the distance of connected vertices and obtain the value associated with the 75% quantile ($Q_{75}$), and we remove all edges connecting vertices, whose distance is greater than three times the $Q_{75}$ value. We denote this as the triangulated graph. We further refine the dummy edges by performing a triangulation between each pair of terminal (high pseudotime) vertices and root (low pseudotime) vertices. This results in SC $= (\mathcal{V}^{(t)}, \mathcal{E}^{(t)}, \mathcal{T}^{(t)})$, where $\mathcal{V}^{(t)}$ is the vertices set, $\mathcal{E}^{(t)}$ is the edge set and $\mathcal{T}^{(t)}$ is the set of triangles. An example of the simplex representation of a differentiation tree is provided in Extended Data Fig. 1b.

The filtering of edges by distance to obtain the triangulated graph is related to PH analysis[35]. To show this, we evaluated how the proposed filtering from the triangulated graph compares to PH analysis of connected components for the mouse embryonic data and simulated data (Extended Data Fig. 4). This comparison suggests that using a filtering step such that we obtain one connected component plus an additional radius (1.2 of the minimum radius to obtain a single component) produces similar results to the previously described filtering of edges. The user can use either approach now in PHLOWER.

**Matrix representation of an SC.** Next, we represent the two-dimensional SC $= (\mathcal{V}^{(t)}, \mathcal{E}^{(t)}, \mathcal{T}^{(t)})$, as incidence matrices or boundary operators $\mathbf{B}_k$ which map $k$-simplices to $(k-1)$-simplices. For example, the first boundary operator $\mathbf{B}_1$ captures the relation between vertices (0-simplices) and edges (1-simplices), and $\mathbf{B}_2$ captures the relation between edges (1-simplices) and triangles (2-simplices)[68].

The boundary matrix on 1-simplices $\mathbf{B}_1$ is defined in equation (8). An entry in $\mathbf{B}_2$ capturing the relationship between an oriented edge $e_i \in \mathcal{E}^{(t)}$ and an oriented triangle $\Delta_q \in \mathcal{T}^{(t)}$ can be defined as shown in equation (11):

$$\mathbf{B}_2[i, q] = \begin{cases} -1 & \text{if } e_i \in \Delta_q \text{ and } e_i \text{ has same orientation as } \Delta_q \\ 1 & \text{if } e_i \in \Delta_q \text{ and } e_i \text{ has opposite orientation as } \Delta_q \\ 0 & \text{if otherwise.} \end{cases} \qquad (11)$$

Check Supplementary Fig. 9 for an example of an SC and its corresponding $\mathbf{B}_1$ and $\mathbf{B}_2$ matrices. There we can find several edges ([1, 2], [2, 3], [2, 4], [2, 5], [3, 4], [4, 5], [4, 6], [4, 7], [6, 7], [7, 8]) and three triangles ([2, 3, 4], [2, 4, 5] and [4, 6, 7]). Entry $B_1[1, 1]$ has a value −1, as the first edge $e_1 = (1, 2)$ leaves vertex 1, while $B_1[2, 1]$ has a value 1, as the first edge [1, 2] enters the vertex 2. Regarding $\mathbf{B}_2$, all entries related to the first edge $e_1 = (1, 2)$ are zero, as there is no triangle associated with it. $B_2[2, 1]$ is equal to one as the direction of $e_2 = (2, 3)$ fits the direction of the first triangle ($\Delta_1 = (2, 3, 4)$). $B_2[3, 1]$ is equal to −1 as the third edge ($e_3 = (2, 4)$) has the opposite direction to $\Delta_1 = (2, 3, 4)$. We refer the reader to ref. 11 for an in-depth characterization of SCs. A similar rationale follows for $\mathbf{B}_2$. Note that for computational simplicity, the orientations of edges and triangles are set with a bookkeeping procedure[11]. That is, vertices are given numerical IDs in order of creation, and edges and triangles are oriented toward vertices with higher ID (Supplementary Fig. 9).

**HL and Hodge decomposition.** The HL is a higher-order generalized form of the graph Laplacian. The $k$-th HL is defined in equation (12):

$$L_k = \mathbf{B}_k^\top \mathbf{B}_k + \mathbf{B}_{k+1} \mathbf{B}_{k+1}^\top \qquad (12)$$

where $\mathbf{B}_k$ is the $k$-th boundary operator.

For $k = 0$, $L_0$ is the same as the graph Laplacian introduced in equation (3), as shown in equation (13):

$$L_0 = \mathbf{0}^\top \mathbf{0} + \mathbf{B}_1 \mathbf{B}_1^\top = \mathbf{D} - \mathbf{A}, \qquad (13)$$

where $\mathbf{D}$ is the diagonal degree matrix of a graph and $\mathbf{A}$ is the adjacency matrix.

Here, we are interested in the first-order HL, as shown in equation (14):

$$L_1 = \mathbf{B}_1^\top \mathbf{B}_1 + \mathbf{B}_2 \mathbf{B}_2^\top. \qquad (14)$$

As in diffusion maps[22], it is preferable to work with the normalized form of the HL, as this provides a random-walk process on the SC[11] as shown in equation (15):

$$\mathcal{L}_1 = \mathbf{D}_2 \mathbf{B}_1^\top \mathbf{D}_1^{-1} \mathbf{B}_1 + \mathbf{B}_2 \mathbf{D}_3 \mathbf{B}_2^\top \mathbf{D}_2^{-1} \qquad (15)$$

where $\mathbf{D}_2$ is the diagonal matrix of (adjusted) degrees of each edge, that is, $\mathbf{D}_2 = \max(\text{diag}(|\mathbf{B}_2|\mathbf{1}), \mathbf{I})$. $\mathbf{D}_1$ is the diagonal matrix of weighted degrees of the vertices according to the edge weights $\mathbf{D}_2$, and $\mathbf{D}_3 = \frac{1}{3}\mathbf{I}$, similar to the standard form of $\mathcal{L}_0$, decomposition hard. To address this, we construct the symmetric form of $\mathcal{L}_1$ with closely related eigenvectors and the same eigenvalues, as shown in equation (16):

$$\mathcal{L}_1^s = \mathbf{D}_2^{-1/2} \mathcal{L}_1 \mathbf{D}_2^{1/2} = \mathbf{D}_2^{1/2} \mathbf{B}_1^\top \mathbf{D}_1^{-1} \mathbf{B}_1 \mathbf{D}_2^{1/2} + \mathbf{D}_2^{-1/2} \mathbf{B}_2 \mathbf{D}_3 \mathbf{B}_2^\top \mathbf{D}_2^{-1/2}. \qquad (16)$$

Next, we perform eigen-decomposition on the symmetric form $\mathcal{L}_1^s$, as shown in equation (17):

$$\mathcal{L}_1^s = Q \Lambda Q^\top \qquad (17)$$

where columns of $Q$ are the eigenvectors and the diagonal elements of the diagonal matrix $\Lambda$ indicate the corresponding eigenvalues. Thus, the decomposition of the normalized form $\mathcal{L}_1$ becomes equation (18):

$$\mathcal{L}_1 = \mathbf{D}_2^{1/2} \mathcal{L}_1^s \mathbf{D}_2^{-1/2} = \mathbf{D}_2^{1/2} Q \Lambda Q^\top \mathbf{D}_2^{-1/2} = \mathbf{U} \Lambda \mathbf{U}^{-1} \qquad (18)$$

where $\mathbf{U} = (\mathbf{u}_1, \cdots, \mathbf{u}_{|\mathcal{E}^{(t)}|})$ is the eigenvector matrix of $\mathcal{L}_1$, $\Lambda = \mathrm{diag}(\lambda_1, \cdots, \lambda_{|\mathcal{E}^{(t)}|})$ are the eigenvalues, and $|\mathcal{E}^{(t)}|$ is the number of edges in the SC. We assume the eigenvectors have been sorted by their corresponding increasing eigenvalues such that $0 \leq \lambda_1 \leq \lambda_2 \leq \cdots \leq \lambda_{|\mathcal{E}^{(t)}|}$. We denote this, according to equation (19),

$$\mathbf{H} := (\mathbf{u}_1, \cdots, \mathbf{u}_h) \qquad (19)$$

to be the matrix containing all the harmonic eigenvectors associated with $\mathcal{L}_1$, that is, all the eigenvectors corresponding to the 0 eigenvalues, where $h$ is the number with eigenvalues being equal to 0 and $\mathbf{H} \in \mathbb{R}^{|\mathcal{E}^{(t)}| \times h}$.

For example, for the embryonic mouse data, we observe that the HL spectra have two zero eigenvalues up to numerical inaccuracies, which define two harmonic eigenvectors. If we plot the harmonic eigenvector values on vertices, we observe that they highlight the two major branches of the triangulated graph (Extended Data Fig. 1c). This is related to the spectral clustering algorithm, where eigenvectors with zero eigenvalues are associated with disconnected components (or clusters) in a graph (Supplementary Fig. 8).

## Trajectory embedding and tree inference

**Trajectory sampling and embedding.** To generate trajectories, we sample paths (or edge flows) in the SC by following edges with positive divergence (or increasing pseudotime). Owing to the sparsity of the SC, we sample paths in $G^{(\mathrm{knn})}$. For this, we perform a preference random walk on graph $G^{(\mathrm{knn})}$. We choose a random starting point from the vertices (cells) with the $m$ lowest pseudotime values. We choose the next vertex randomly by considering the divergence values ($w^s$). Only positive divergences (increase in pseudotime) are considered. We stop when no further positive potential is available. Next, we return to the SC, estimating the shortest paths between vertices in case-sampled edges (from $G^{(\mathrm{knn})}$) are not present in the SC. We define the embedding $\mathbf{f} \in \mathbb{R}^{|\mathcal{E}^{(t)}|}$ of a path on the SC into the edge-flow space as shown in equation (20):

$$f[i,j] = \begin{cases} 1 & \text{if edge}(i,j) \text{ is traversed} \\ -1 & \text{if edge}(j,i) \text{ is traversed} \\ 0 & \text{otherwise} \end{cases} \qquad (20)$$

The random walk is repeated $n$ times. This provides us with a path matrix $\mathbf{F}^{(t)} \in \mathbb{R}^{|\mathcal{E}^{(t)}| \times n}$, where $|\mathcal{E}^{(t)}|$ is the number of edges in the SC. See Extended Data Fig. 1d for examples of sampled trajectories or edge flows. We next project these paths $\mathbf{F}^{(t)}$ onto harmonic space to estimate a trajectory embedding as shown in equation (21):

$$\mathbf{H}^{(t)} = \mathbf{H}^\top \mathbf{F}^{(t)} \qquad (21)$$

where $\mathbf{H} \in \mathbb{R}^{|\mathcal{E}^{(t)}| \times h}$ are the harmonic eigenvectors defined in equation (19). PHLOWER next performs clustering on the trajectory embedding $\mathbf{H}^{(t)}$ with DBSCAN[69] to group the paths into major differentiation trajectories. For the MEF data, we observe that the trajectory embedding (or trajectory map) reveals two clusters (Extended Data Fig. 1e) associated with the neuronal and myocyte differentiation.

**Cumulative trajectory embedding and tree inference.** The path representation presented in equation (20) does not keep the time step of an edge visit. Thus, we also define a traversed edge-flow (traversed path) matrix $\hat{\mathbf{f}} \in \mathbb{R}^{|\mathcal{E}^{(t)}| \times S}$ to record the edge visits for each time step individually, that is, as shown in equation (22):

$$\hat{f}[i,j,s] = \begin{cases} 1 & \text{if edge}(i,j) \text{ is traversed in step } s \\ -1 & \text{if edge}(j,i) \text{ is traversed in step } s \\ 0 & \text{otherwise} \end{cases} \qquad (22)$$

where $S$ is the length of the trajectory, $1 \leq s \leq S$ is the number of the step and $|\mathcal{E}^{(t)}|$ is the number of edges in the graph SC. As we have $n$ trajectories, we will have $n$ traversed edge-flow matrices $\{\hat{\mathbf{f}}_1, \hat{\mathbf{f}}_2, \cdots, \hat{\mathbf{f}}_n\}$.

We use cumulative trajectory embedding to represent paths and to detect major trajectory groups and branching points. For a path $\hat{\mathbf{f}}$, we can obtain a point associated with every step $s$ in this cumulative (harmonic) trajectory embedding space as shown in equation (23):

$$\mathbf{v}_s = \mathbf{H}^\top \sum_{i=1}^{s} \hat{\mathbf{f}}_{,i} \qquad (23)$$

where $\hat{\mathbf{f}}_{,i} \in \mathbb{R}^{|\mathcal{E}^{(t)}|}$ is a signed indicator vector for the edge traversed in step $i$, $\mathbf{v}_s \in \mathbb{R}^h$ is the cumulative trajectory embedding of path $\hat{\mathbf{f}}$ in step $s$ in harmonic coordinates and $h$ is the number of harmonic eigenvectors with zero eigenvalues. This is computed for all $1 \leq s \leq S$, which defines a vector $\mathbf{v} = \{\mathbf{v}_1, \ldots, \mathbf{v}_S\}$ for every path. Intuitively, the entries of $\mathbf{v}$ describe a trajectory in the harmonic edge-flow space that starts at 0 and ends in the harmonic trajectory embedding of the entire path.

As observed in Extended Data Fig. 1e, these vectors are low-dimensional representations of paths in the cumulative trajectory embedding. By coloring paths from distinct groups with distinct colors, we can recognize branching-point events, branches shared by trajectory groups and terminal branches. Note also that if we consider only the final entry $\mathbf{v}_s$ for every path, we obtain the same result as in the previously described trajectory embedding.

Recall that we have performed the DBSCAN clustering method to cluster the $n$ paths into $m$ groups $\{g_1, g_2, \cdots g_m\}$ on the trajectory embedding defined in equation (21). PHLOWER next uses a procedure schematized in Supplementary Fig. 10 to find the differentiation tree structure. First, it estimates pseudotime values for every edge, that is, the average pseudotime $u^s$ from vertices associated with the edge. It next bins all edges by considering their pseudotime, that is, it selects the trajectory group with the lowest pseudotime and splits its edges in $p$ bins (Supplementary Fig. 10b). The same range of pseudotime is used to bin all trajectory groups and bins are indexed in increasing pseudotime (Supplementary Fig. 10c). After binning of edges for each group, PHLOWER next finds the branching points for all group pairs by comparing the distance of edges within the bin versus the distance of edges between the bins for a given bin index $i$.

More formally, for groups $g_i$ and $g_j$ and bin $k$, their corresponding edges in cumulative space are defined as set $\mathbf{V}_i^k$, $\mathbf{V}_j^k$. We then estimate the average edge coordinate per bin to serve as backbones for every group, that is, as shown in equation (24):

$$\overline{b_i^k} = \frac{1}{M} \sum_{v \in \mathbf{V}_i^k} v, \qquad (24)$$

where $M = |\mathbf{V}_i^k|$ is the number of edges in the bin. We also consider the average distance between edges in a bin to have an estimate of the compactness of edges in a bin and trajectory, as shown in equation (25):

$$\overline{\sigma_i^k} = \frac{1}{M^2} \sum_{u \in \mathbf{V}_i^k} \sum_{v \in \mathbf{V}_i^k} \|u - v\|_2. \qquad (25)$$

Finally, we calculate the distance between two groups $g_i$ and $g_j$ in time bin $k$, as shown in equation (26):

$$d(i,j,k) = \left\| \frac{\overline{b_i^k}}{\overline{\sigma_i^k}} - \frac{\overline{b_j^k}}{\overline{\sigma_j^k}} \right\|_2. \qquad (26)$$

For every pair of groups, PHLOWER finds a unique branching point by traversing bins in decreasing order and finding the first bin such that $d(i,j,k) < \sigma$ (as default 1).

This is repeated for all pairs of groups. The tree is finally built in a bottom-up manner. PHLOWER first considers branching points with the highest index and builds a sub-tree by merging the two trajectory groups at hand. This is repeated until all branching points are considered (Supplementary Fig. 10d,e). PHLOWER finally defines the leaves and the root by finding more extreme points, and edges with lowest and highest pseudotime in a trajectory group. These are the so-called milestones (branching points, root and end points) needed for evaluation by Dynverse[8]. Trajectories are redefined as branches, that is, part of the trajectories between two milestones. We also allocate cells (vertices) to these branches. For this, we consider the location of all edges associated with a vertex and use the mean value in the cumulative trajectory embedding space. This is used to allocate cells to branches and to find the distance between respective milestones. Moreover, we provide this information for STREAM[5] for visualization as a STREAM tree.

**Regulator and marker selection.** We use the cumulative trajectory space and statistical tests to find markers and regulators associated with trajectories. To compare two final branches, PHLOWER selects all cells associated within particular areas of the branches, that is, the highest 50% of the bins. Note that every cell can be visited by several edges. To consider this important information, we multiply the expression count vector of each cell by the number of visits. We then perform a statistical test (default is the $t$-test from Scanpy[70]). In case we are interested in comparing sub-trees with several branches, PHLOWER first merges the sub-tree as a single branch, before the selection of the bin.

In the case of multimodal data (RNA and ATAC), PHLOWER also has a module to find regulators. First, we estimate a TF activity score per cell using chromVar[71]. We then use the previously described test to find branch-specific regulators. Since TF activity cannot be discerned from TFs with similar motifs, we only consider genes whose TF activity is similar to the expression pattern at the selected branch as in ref. 45. We first calculate the average expression/TF activity of cells of bins around the branch of interest. We next smooth the gene expression using convolution (numpy.convolve)). We then estimate the correlation between gene expression and TF activity and use this to sort branch-specific regulators.

### Materials

**Synthetic datasets.** To test the power of PHLOWER in the detection of multi-branching trees, we generate ten simulated datasets using DLA trees[22] as proposed in PHATE[14]. For this, we use a Gaussian noise parameter of 5 and vary the parameter $n\_branch$, which controls the number of branches in the trees, from 3 to 12. This generated trees with 5 to 18 branches. For all data, we generate 3,000 points and 100 features. Of note, we use the same data as evaluated in PHATE as data with 10 branches. Next, we run PHATE to visualize the branches, with which we can construct ground truth for Dynverse benchmarking in the future.

Next, we reformat this data to be used within the trajectory benchmarking framework Dynverse[8] by using Dynwrap. In short, Dynverse needs detailed information of the branches, branch points and association of cells within a branch. To do so, we need to find the branching points of the DLA tree. First, we perform PHATE on the DLA high-dimensional data to get embedding with two dimensions. We only consider embeddings, where the tree structure is preserved. Next, we find the branching points by finding two nearest neighbors of two branches. With these branching points, we created the branch backbones needed by Dynverse. We calculate the association of each data point to a branching point (milestone percentage) by calculating the Euclidean distance between the points related to each branch.

**Real scRNA-seq datasets.** To evaluate the performance of PHLOWER on real datasets, we selected 33 real datasets from the Dynverse benchmarking dataset, including 4 gold-standard and 29 silver-standard datasets with known ground-truth tree structures. Specifically, we only consider data with a single root and at least three branches. These are classified as bifurcation, multifurcation and tree structures as in Dynverse (Supplementary Table 1). For this, we inspected the code from https://github.com/dynverse/dynbenchmark/ and ran the script to download all the necessary data.

### Benchmarking evaluation

Dynverse includes wrappers for several trajectory methods. We explore the following methods in our evaluation: PAGA tree[4], Monocle3 (ref. 23), Slingshot[6], Slice[26], TSCAN[29], Slicer[72], MST[30] and Elpigraph[31]. We did not included Slicer in the final benchmarking as it obtained poor results. We expand this by including a wrapper for STREAM[5] and PHLOWER. Note that some tools (Palantir and MIRA[7,73]) do not infer differentiation trees and can, therefore, not be evaluated here. We first calculate the distance of each cell to all branches. Next, we assign the cell to the nearest branch and use the distance ratio to associate cells to milestones. Then, we use Dyneval to measure quality metrics for the generated trajectories. As in ref. 8, we mainly focus on HIM distance to measure the tree structure similarity; $cor_{dist}$ to measure the correlation between the cell geodesic distances within predicted and true branches; and $F_{1,branches}$ and $F_{1,milestones}$ to measure the accuracy of a cell assigned to the correct branches and the correct milestone (bifurcation points). To obtain a final accuracy score, we used a procedure as described in Dynverse, which is based on the average of the four previous statistics per dataset.

### Statistics and reproducibility

Benchmarking analyses were evaluated with the Friedman–Nemenyi post hoc test[74]. First, Friedman's test[75] was performed to compare the average ranks of the methods across all datasets. Nemenyi's post hoc test[76] was performed for pairwise multiple comparisons. For imaging experiments, the number of samples for each group was chosen on the basis of the expected levels of variation and consistency. The depicted immunofluorescence graphs are representative. All imaging experiments were performed at least three times, and all repeats were successful.

### Kidney organoids

**Ethical statement.** Permission for the creation and use of iPS cells in this study was obtained from the ethical commission at RWTH Aachen University Hospital (approval number EK23-193).

*Cell culture.* For the generation of the human iPS cell-353 line, erythroblasts from a healthy male volunteer were reprogrammed using the CytoTune-iPS 2.0 Sendai Reprogramming Kit (Thermo Fisher) according to the manufacturer's protocol. In short, erythroblasts were transduced using the CytoTune-iPS 2.0 Sendai Reprogramming Kit and re-seeded on plates with inactivated MEFs to support the growth and pluripotency of iPS cells. Emerging iPS cell colonies were picked, expanded and assessed for activation of stem cell markers to confirm pluripotency.

*Generation of human iPS cell-derived kidney organoids.* For the generation of human iPS cell-derived kidney organoids, iPS cells were seeded using a density of 20,000 cells per cm² on Geltrex (Thermo Fisher)-coated six-well plates (Greiner Bio-One). The differentiation protocol was based on our previous work[36]. In brief, differentiation toward ureteric bud-like and metanephric mesenchyme lineage was initiated using CHIR-99021 (10 μM, BioTechne) in STEM diff APEL 2 medium (StemCell Technologies) for 3 and 5 days, respectively. Next, the medium was replaced sequentially for APEL 2 supplemented with fibroblast growth factor 9 (200 ng ml⁻¹, R&D systems) and heparin (1 μg ml⁻¹, Sigma-Aldrich) up to day 7. On day 7, differentiated cells were trypsinized and mixed in a ratio of one part 3-day CHIR-differentiated cells and two parts 5-day CHIR-differentiated cells to stimulate

cross-talk between both lineages to boost segmented nephrogenesis. To generate cell pellets, 300,000 cells per 1.5-ml tube were aliquoted from the cell mix and centrifuged three times at 300g for 3 min changing position by 180° per cycle. Cell pellets were plated on Costar Transwell filters (type 3450, Corning, Sigma-Aldrich), followed by a 1-h CHIR pulse (5 μM) in APEL 2 medium added to the basolateral compartment. Next, medium was replaced for APEL 2 medium supplemented with fibroblast growth factor 9 and heparin for an additional 5 days and the entire three-dimensional organoid culture was performed using the air–liquid interface. After 5 days of organoid culture, APEL 2 medium was supplemented with human epidermal growth factor (10 ng ml⁻¹, Sigma-Aldrich). Medium was replaced every other day for an additional 13 days.

*Silencing of ZIC2, RFX4 and PAX3 during stem cell-derived kidney organoid development.* Kidney organoids (at least $N = 3$ per condition per experiment, two independent experiments) were transfected using ON-TARGETplus SMARTpool siRNA's ZIC2 (L-017505-00-0005), RFX4 (L-013577-00-0005) and PAX3 (L-012399-00-0005, work concentration 25 nM, Horizon Discovery) and DharmaFECT Transfection reagent 1 (0.5% vol/vol) from day 7+5 (intermediate mesoderm stage) onward to the end of the protocol until day 7+18. Organoids were refreshed every other day.

*RNA extraction, cDNA synthesis and qPCR.* RNA from organoids was extracted using the PureLink RNA mini kit (Thermo Fisher) according to the manufacturer's protocol. RNA was stored at −80 °C until further processing. cDNA synthesis was performed using 200 ng RNA as input, using the iScript cDNA synthesis kit (Bio-Rad) according to the manufacturer's protocol. The mRNA expression was quantified by performing a semi-quantitative real-time PCR using PowerUp SYBR Mastermix (Applied Biosystems) and primers targeting human *RFX4*, *ZIC2* and *PAX3* genes (hZIC2_For AAAGGACCCACACAGGGGAGA, hZIC2_Rev GACGTGCATGTGCTTCTTCCT, hRFX4_For TGGGAAGAGCATGCATTGTG, hRFX4_Rev TCTTTCAATCCAGCTCTCTGTGG, hPAX3_For GCAGTATGGACAAAGTGCCT, hPAX3_Rev CAGGGCCAGTTTTAGCTCCA). After correction with the corresponding human *GAPD* gene (GAPDH_For GAAGGTGAAGGTCGGAGTCA, GAPDH_Rev TGGACTCCACGACGTACTCA), gene expression levels were plotted as fold change compared to control. Data plotting and statistical analysis were performed using GraphPad Prism (version 10.0.3).

*Harvesting kidney organoids for 10x genomics NEXT GEM multiome pipeline.* To dissect nephrogenic differentiation trajectories in kidney organoids using the multiome pipeline, organoids were collected at different time points during culture. The following differentiation stages were processed: day 7 (cells harvested from the two-dimensional cell layer, primitive streak - intermediate mesoderm stage), organoids day 7+5 (day 12), day 7+12 (day 19) and day 7+18 (day 25). These kidney organoid stages represent nephrogenesis ranging from intermediate mesoderm (day 7+5) toward metanephric mesenchyme and ureteric bud-like lineages (day 7+12) that result into nephron-like structures embedded by a (progenitor) stromal compartment at the end of the differentiation protocol (day 7+18). Kidney organoids ($N = 4$ per time point) were collected on the respective dates (day 7+5, day 7+12 and day 7+18) by cutting single organoids out of the Transwell filter in a sterile flow hood using a scalpel. Organoids were washed with 5 ml PBS per filter at room temperature three times. Afterwards, single organoids were cut from the Transwell filter with the organoids still being attached to the membrane, transferred to 1.7-ml cryovials (Greiner Bio-One) and subsequently snap frozen and stored at −80 °C until nuclei isolation.

*Single-nuclei isolation from kidney organoids.* Snap-frozen kidney organoids were thawed in PBS and crushed using a glass tube and douncer (Duran Wheaton Kimble Life Sciences). After passing the single-cell suspension through a 40-μm cell strainer (Greiner Bio-One), the suspension was centrifuged at 4 °C and 300g for 5 min. Subsequently, the supernatant was discarded and the cell pellet was resuspended in Nuc101 cell lysis buffer (Thermo Fisher), supplemented with RNase and protease inhibitors (Recombinant RNase Inhibitor and Superase RNase Inhibitor, Thermo Fisher, and cOmplete Protease Inhibitor, Roche), incubated for 30 s and centrifuged at 4 °C and 500g for 5 min. After discarding the supernatant, the nuclei were carefully resuspended in PBS containing 1% (vol/vol) Ultra-Pure bovine serum albumin (BSA; Invitrogen Ambion, Thermo Fisher) and Protector RNAse inhibitor (Sigma-Aldrich). Single nuclei were counted using Trypan blue (Thermo Fisher) and prepared for 10x genomics ChromiumNextGEM Multiome pipeline v1 according to the manufacturer's guidelines (https://cdn.10xgenomics.com/image/upload/v1666737555/support-documents/CG000338_ChromiumNextGEM_Multiome_ATAC_GEX_User_Guide_RevF.pdf).

*Formalin-fixed paraffin-embedded tissue of human iPS cell-derived kidney organoids.* iPS cell-derived kidney organoids were cut from the Transwell filter and fixated in 4% (vol/vol) formalin on ice for 20 min. Fixed iPS cell-derived kidney organoids were stripped of the filter membrane using a scalpel. Each single organoid was embedded using 2.25% (wt/vol) agarose gel (Thermo Fisher). After embedding for 5 min at 4 °C, the iPS cell-derived kidney organoids were transferred to embedding cassettes and paraffinized. After paraffinization, iPS cell-derived kidney organoids were cut at a thickness of 4 μm using a rotary microtome (Microm HM355 S, GMI) and mounted on FLEX IHC microscope slides (DAKO, Agilent Technologies) for immunofluorescence staining or Xenium array slides for spatial analysis.

*Immunofluorescence staining.* Slides were deparaffinized using a series of xylol (2×) and 100% (vol/vol) ethanol (3×), followed by antigen retrieval by boiling slides in Tris-buffered EDTA (VWR Chemicals) for 20 min. Primary (1:100 dilution) and secondary (1:200 dilution) antibodies were diluted in PBS containing 1% (vol/vol) BSA (Sigma-Aldrich). Primary antibodies (NPHS1, AF 4269-SP, RD Systems, Vimentin, ab92547, Abcam, E-cadherin, 610405, BD Biosciences) were incubated overnight at 4 °C, and secondary antibodies were incubated at room temperature for 2 h (donkey anti-sheep IgG (H+L) Alexa Fluor 647 (Thermo Fisher), donkey anti-rabbit IgG (H+L) Alexa Fluor 488 (Thermo Fisher), donkey anti-mouse IgG (H+L) Alexa Fluor 488, DAPI (300 nM, 4′,6-diamidine-2′ phenylindole dihydrochloride, Merck). After each antibody incubation, slides were washed three times in PBS for 5 min. Slides were mounted using Fluoromount-G (Southern Biotech, SanBio). Images were captured using a Zeiss LSM 980 confocal microscope. Image analysis was performed using Fiji v2.14, and conditions were normalized to control. Data plotting and statistical analysis were performed using GraphPad Prism (version 10.0.3).

**Computational analysis of kidney organoids.** After read mapping using the cellranger-arc tool (version 1.0.1), we filter the low-quality cells using information both from scRNA and scATAC reads. We first import scRNA to Seurat[77] to get the scRNA metrics for filtering. We next import the fragments into ArchR[78] to get the scATAC metrics for filtering. With the information above, we retain cells with barcodes in both scRNA and scATAC count matrices. Next, we filter the cells using threshold nFeatureRNA > 400 and nCountRNA <400,000 and percent. mito > 5 and scATAC using thresholds in ArchR that minTSS = 6 and minFrags = 2,500 and maxFrags = $1 × 10^5$. We next perform preprocessing to scRNA using Seurat. To do this, we first normalize the scRNA data by calling NormalizeData with the default parameters. Next, we find top variable features with parameter selection.method = 'vst'. Then, we scale the data by regressing out the cell cycle and mitochondrial effect. We next run PCA with 50 principal components (PCs). To remove the batch effects, we next run harmony[37] to integrate the four samples.

For scATAC data, we use ArchR to perform the preprocessing. We first create Arrow files using the aforementioned filter threshold by calling createArrowFiles; the tile matrices are created directly from the fragment files. Next, we add doublet scores by calling addDoubletScores for each sample followed by the filterDoublets function to remove doublets with default parameters. Next, we run addIterativeLSI on the tile matrices to add dimensional reductions. A batch correction is also called using Harmony[37] with the addHarmony function. To obtain a uniform dimension reduction for the downstream analysis, we next run MOJITOO[9] to integrate the scRNA and scATAC data with default parameters.

MOJITOO embedding was used as input for PHLOWER. An important parameter is the indication of the root cells. For this, we performed a clustering analysis using Seurat[77] with resolution = 2.5 on the MOJITOO embedding space. The root cells were defined by clusters predominantly present in day 7 and with the expression of mesoderm markers (TBXT, MESP1, KDR5). PHLOWER found 28 dimensions with zero eigenvalues and clustering analysis of the trajectory space detected 16 groups. We removed four main trajectory outliers (where less than 0.5% of samples are inside the cluster) and one trajectory, where cells had a low pseudotime values (for example, they did not differentiate). Finally, some clusters had the same end time points. Therefore, we kept the one with the largest number of trajectories. This resulted in the nine final trajectories found by PHLOWER.

**Computational analysis of Xenium experiments of kidney organoids.** To select markers, we used the phlower.tl.tree_mbranch_markers function to identify markers for each main branch (stromal, neuronal, tubular and podocytes), focusing on cells with the highest pseudotime according to PHLOWER (top 50%). Next, we further selected markers with expression specific to sub-branches (tubular, podocytes, stromal 1, stromal 2–4, neuron 1 and neuron 2–3). For this, we retained only genes with an adjusted $P$ value < 0.05 and log fold change (FC) > 1.5 for the main branch markers. Next, we only considered protein-coding genes according to Ensembl (version GRCh38.104). We then scaled the logFC values of both main branches and sub-branches to a 0–1 range and calculated the mean normalized logFC for each branch and its sub-branches. We selected the top 12 genes for each sub-branch based on the mean normalized logFC. For TF selection, we used TF–gene correlation along each sub-branch to identify five TFs for each main branch using scMEGA analysis (Extended Data Fig. 9). Additionally, we included literature-based cell-type markers reported in Extended Data Fig. 7, which supported the annotation of the multiome single-cell data. In total, we identified 100 genes for the Xenium experiment (Supplementary Table 2). The panel primers were designed with the Xenium Panel Designer tool provided by 10x Genomics.

For the data analysis, we used Xenium Explore3 to inspect each region and remove those that did not show differentiation and remove stromal cell patches around organoid edges. This process left us with two scrambled siRNA regions, one siRNA region, one day-19 region and two day-25 regions. Note that multiple regions correspond to distinct sections of the same organoid. Next, we used Space Ranger with default parameters and loaded each Xenium sample using Seurat (version 5.0.2) with the LoadXenium function and removed cells based on the organoid masks. We integrated all cells using Seurat's merge function and filtered out cells with zero nCount_Xenium. We next applied SCTransform for data normalization, PCA for dimensionality reduction and Louvain clustering (50 PCs and resolution = 0.3), and estimated uniform manifold approximation and projection for visualization. Next, we annotated the clusters by examining the gene expression of known cell-type markers. We excluded five clusters with 33,396 cells owing to low nFeature_Xenium (<20) and nCount_Xenium (<60). This potentially represents cells whose markers are not present in our panel.

For the day-19 and day-25 regions, we performed trajectory inference using PHLOWER using a subset of the cells (2,000 per cluster). We used mesoderm cells as the root for the trajectory inference and identified three end branches. Finally, we integrated PHLOWER with STREAM for visualization.

### Pancreas and neurogenesis scRNA-seq data

We applied PHLOWER and competing approaches to pancreatic endocrinogenesis data (3,696 cells)[32] and the data on developing mouse hippocampus (neurogenesis, 18,213 cells)[33], both of which were used in scVelo[34]. Specifically, we extracted AnnData using the scVelo package via scvelo.datasets.pancreas and scvelo.datasets.dentategyrus_lamanno. For both datasets, we applied the Scanpy pipeline including filtering, feature selection, normalization and PCA with default parameters. See tutorials for details on how PHLOWER was executed in these datasets.

For competing approaches, these methods were executed with Dynverse. Basically, we first use R to load the h5ad file with anndata::read_h5ad and create a SeuratObject from the count matrix. We then perform normalization and identify the top 2,000 variable features to obtain the log-normalized expression matrix. Next, we use dynwrap::wrap_expression to structure a Dynverse dataset object with count and expression data. The start_id is set as the first 'Ductal' and 'nIPC' cell ID for pancreas and neurogenesis, and group_id is assigned based on cell types. The dataset object is then saved as an R object, enabling trajectory inference with dynwrap::infer_trajectory. Finally, we visualize the inferred trajectories using dynplot::plot_dimred.

### Reporting summary

Further information on research design is available in the Nature Portfolio Reporting Summary linked to this article.

## Data availability

Data objects with the kidney organoid (multiome and Xenium) and the benchmarking data have been deposited in Zenodo via https://doi.org/10.5281/zenodo.13860460 (ref. 79). Single-cell multimodal data and spatial transcriptomics have been deposited in the Gene Expression Omnibus (GEO) under the accessions GSE302266 and GSE302264. Source data are provided with this paper.

## Code availability

Code, documentation and examples for running analysis of this paper are available on GitHub via https://github.com/CostaLab/phlower/ and readthedocs via https://phlower.readthedocs.io/en/latest/.

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

## Acknowledgements

We thank the IZKF Aachen Genomics Core facility for sequencing experiments. This project has been funded by the German Research Foundation (DFG; 3888802535, CRU344-417911533, CRU344-4288578857858, CRU5011-445703531, CRU5011-445703531, SFBTRR219 322900939, Emmy Noether EN-459969915 and Research Training Group 2236 UnRAVeL), by the Consortia E:MED Fibromap, CureFib, Graphs4Patients and AgedHeart funded by the German Ministry of Science, Technology and Space (BMFTR). This work was further supported by grants from the European Research Council (ERC-COG 101043403, ERC-PoC 101138549, ERC-StG-101040726, ERC-StG-101039827 HIGH-HOPeS), the Dutch Kidney Foundation (DKF), TASKFORCE EP1805, the NWO VIDI 09150172010072, Else Kroener Fresenius Foundation (EKFS) and the Aventis Foundation. Views and opinions expressed are those of the authors only and do not necessarily reflect those of the European Union or the European Research Council Executive Agency. Neither the European Union nor the granting authority can be held responsible for them.

## Author contributions

M.C., I.C., V.G. and M.S. conceived computational methods, while J.J., K.R., C.K. and R.K. conceived organoids, wet-lab and sequencing experiments. M.C. wrote the code and performed computational analysis, except where otherwise noted. V.G., J.N. and M.G. supported the implementation of PHLOWER. Z.L. performed the analysis of the single-cell multimodal data, while P.K. supported the analysis of the Xenium data. J.J. and K.R. performed the organoid experiments and J.J. performed knockdown validations. All authors edited, reviewed and approved the final manuscript.

## Funding

## Competing interests

R.K. is founder and board member of Sequantrix, is a member of the scientific advisory board of Hybridize Therapeutics, received honoraria from Bayer, Chugai, Pfizer, Roche, Genentech, Lilly and GSK, and received research funding from Travere Therapeutics, Galapagos, Novo Nordisk and Ask Bio. The other authors declare no competing interests.

## Additional information

**Extended data** is available for this paper at https://doi.org/10.1038/s41592-025-02870-5.

**Correspondence and requests for materials** should be addressed to Rafael Kramann or Ivan G. Costa.

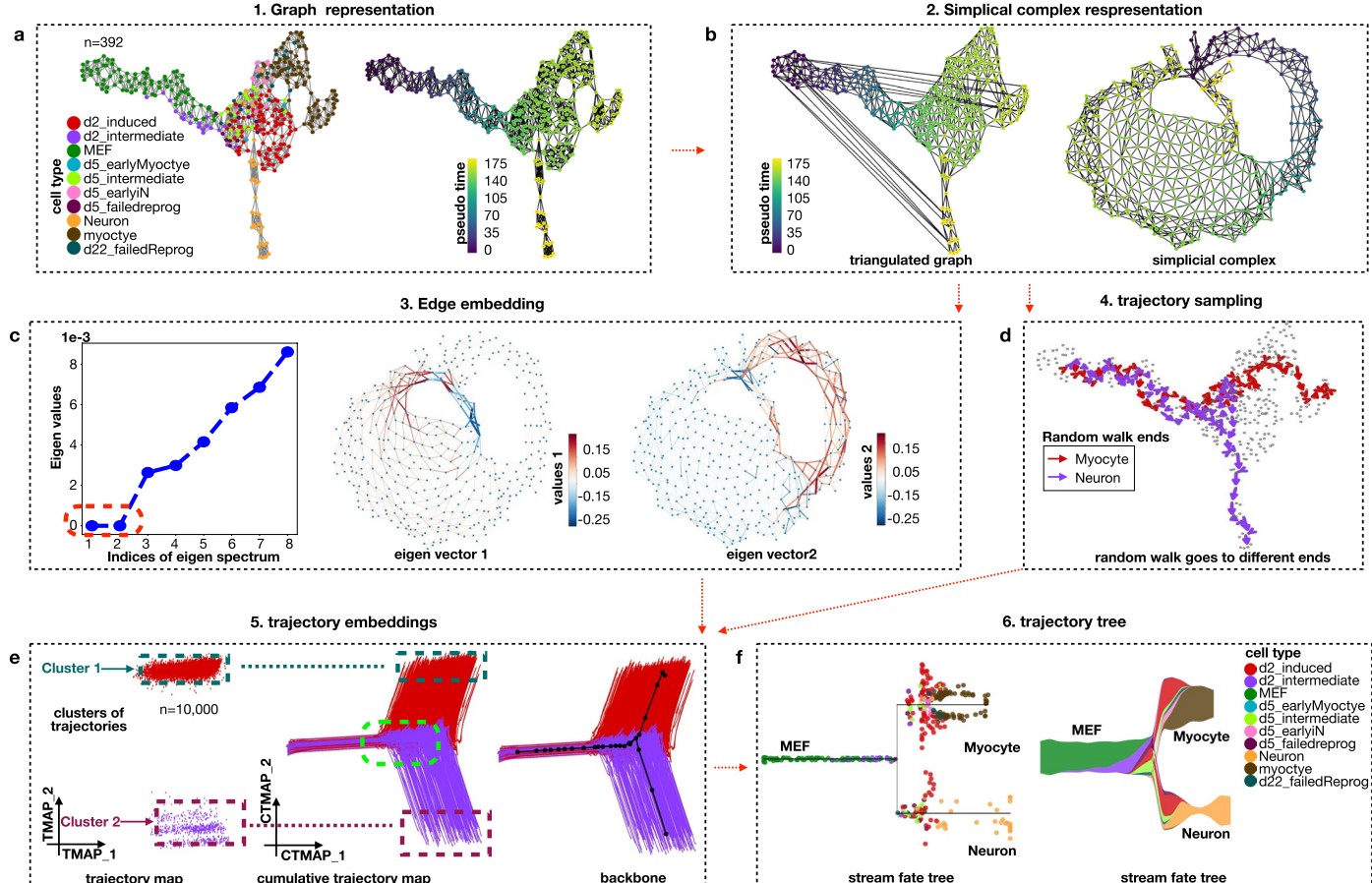

**Extended Data Fig. 1 | Detailed steps of PHLOWER on a single cell data with mouse embryonic fibroblasts towards neurons and myocytes.**
**a**, First, PHLOWER estimates a graph representation using kernels and stress majorization layout[62]. Colors correspond to cluster labels as defined in[21]. This information (labels) is not used by PHLOWER algorithms. **b**, Next, a pseudo-time is estimated using the zero-order Laplacian and using MEF cluster as root. The definition of a root is the only supervision required in PHLOWER. Edges between cells are obtained via Delaunay triangulation (left). To create holes in the simplicial complex, PHLOWER performs a trick by including edges between high pseudo-time towards low pseudo time vertices (middle). If we re-run the graph layout, two holes are clearly visible. **c**, PHLOWER next computes the harmonic eigenvectors of the Hodge Laplacian of the simplicial complex. This indicates two harmonic eigenvectors, that is eigenvectors with zero eigenvalues. By plotting harmonic eigenvector's values on the edges, we observe that the values of the

first harmonic eigenvector discriminates edges in the neuronal branch vs. others, while the values of the second harmonic eigenvector discriminates the myocyte related cells from others. **d**, Next, PHLOWER generates trajectories by random walk on the simplicial complex representation to obtain edge-flow vectors. **e**, Finally, PHLOWER creates a trajectory embeddings (Eq. (21)) by a dot product of the edge flows with the harmonic eigenvectors. We observe two clear clusters in the trajectory map (left), which are detected by providing this embedding as input for DBSCAN[69]. We can similarly create a cumulative trajectory map (Eq. (23)) by creating edge flows of trajectories by considering only the first one, two, three and so on first differentiation events. PHLOWER uses this space to delineate the backbones of differentiation trajectories and to detect branching points events (in green, right panel). **f**, PHLOWER outputs a differentiation tree, pseudo-time values and an allocation of cells to positions in the branches. PHLOWER makes use of STREAM to visualise the final trajectories.

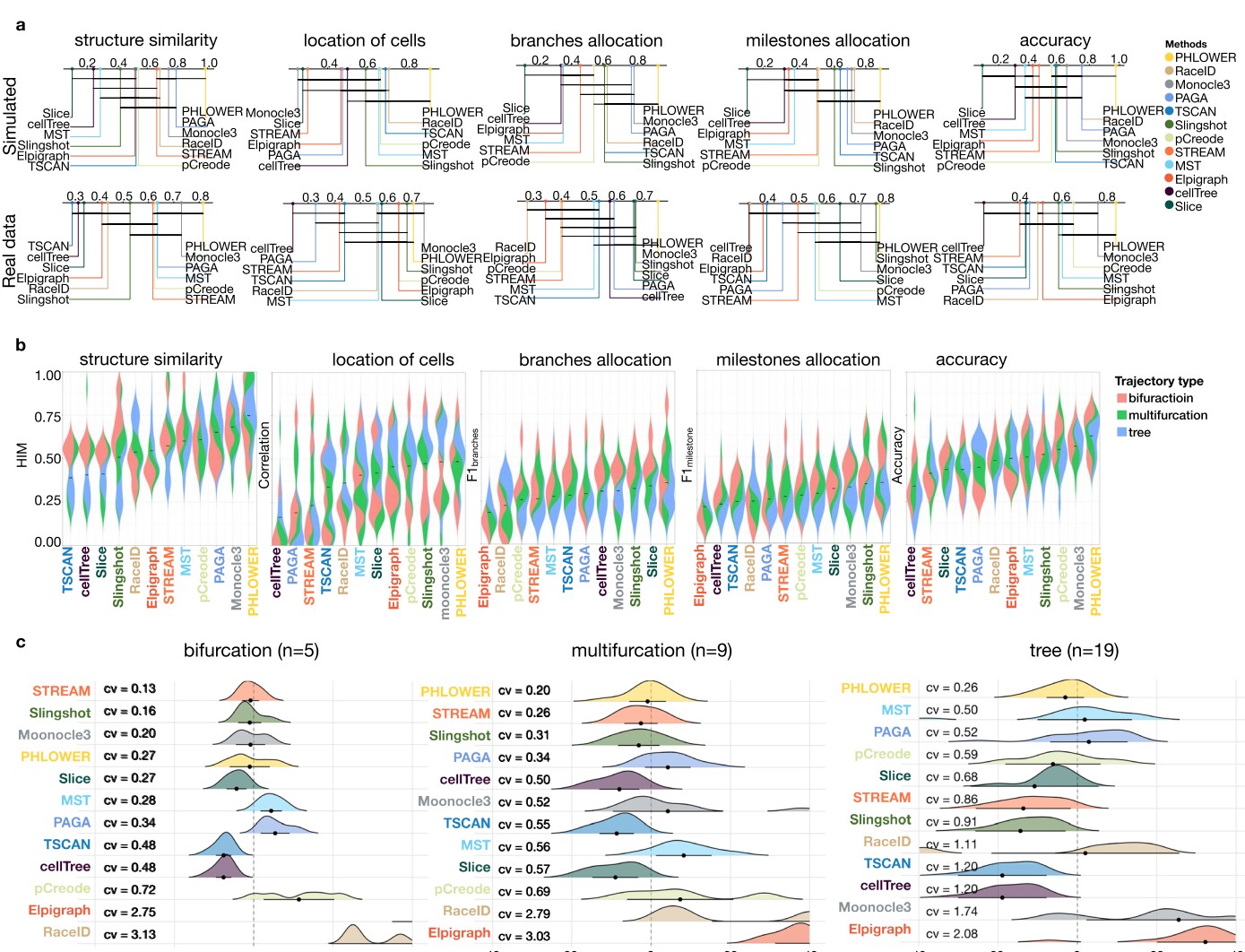

**Extended Data Fig. 2 | Ranking of Benchmarked methods. a**, We show the mean rankings of the methods based on the HIM, Correlation, F1 branches and F1 milestones for all evaluated methods in the simulate data set (based on results from Fig. 2a) and scRNA-seq data sets (based on results from Fig. 2b). Higher ranking values indicate the best performer. Bars indicate methods with similar performance in accordance to the Friedman-Nemeniy test. **b**, HIM, correlation, F1 branches, F1 milestones and accuracy score (y-axis) vs. methods (x-axis) for 33 real scRNA-seq datasets. Methods are sorted by increasing mean (black line), while distinct color indicates the distribution of score per type of structure: bifurcation, multifurcation and trees. While some approaches perform

comparatively similar with all type of structures, some approaches perform relatively better for more complex structures, that is PAGA has higher HIM scores for trees, while slighshot performs best for simpler bifurcation. **c**, Distributions of topology size differences between predicted and reference structures. Positive values indicate that the predicted topologies are more complex than the references, while negative values indicate simpler trees were estimated. The dots represent the mean values, and the lines indicate the standard deviations. The best approach should have the lowest absolute coefficient of variation (cv=mean/std). Note that this statistic does not evaluate the topological similarity of the tree, which is indicated by the HIM statistics.

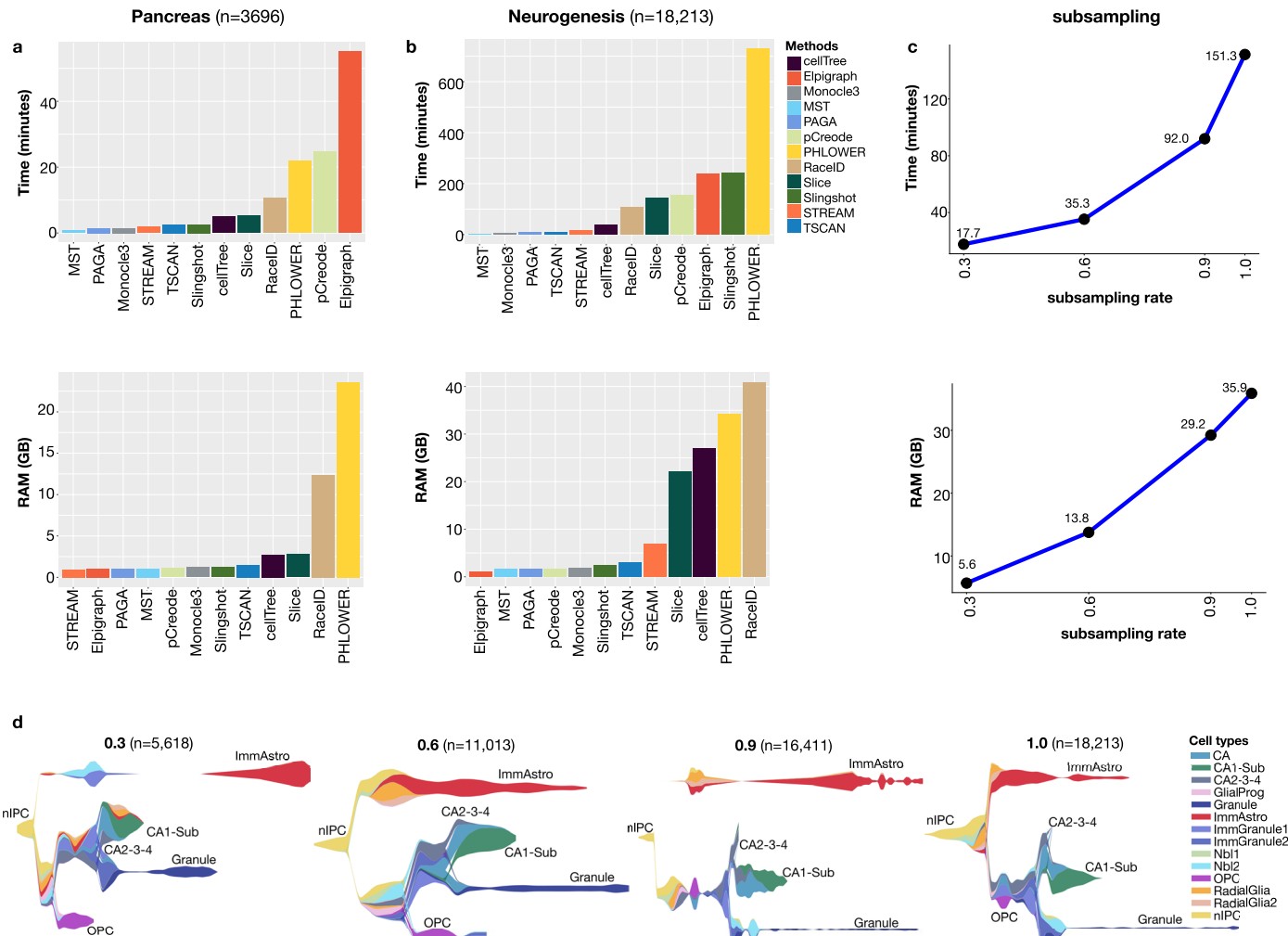

**Extended Data Fig. 3 | Time and Memory Benchmarking. a**, We show the time (left) and memory (right) requirements for pancreas progenitor (3.7k cells) and **b**, neurogenesis (18k cells) data. The profiling was performed with the package memory-profiler (0.61.0) on a computer with an Intel i5 10400 processor (12 threads), 64GB RAM, running Linux Mint OS 21.1. **c**, We show the time and memory requirements of PHLOWER for the neurogenesis data (18k cells) after subsampling by using between 30% to 90% of cells. The use of half of the data provides a 8.6x speed up and requires 1/6 of the memory when compared to using all cells. Experiments were executed on a high perfomance computing node (AMD EPYC 7543 50/128 Cores 2.345G, 1024GB RAM, Rocky Linux 8.9.). Note that this provides an speed up of 4x compared to experiments in a-b, which needed to be executed in a normal desktop computer due to super user requirements of Dynverse. **d**, PHLOWER stream trees obtained for the distinct sub-sampling procedures.

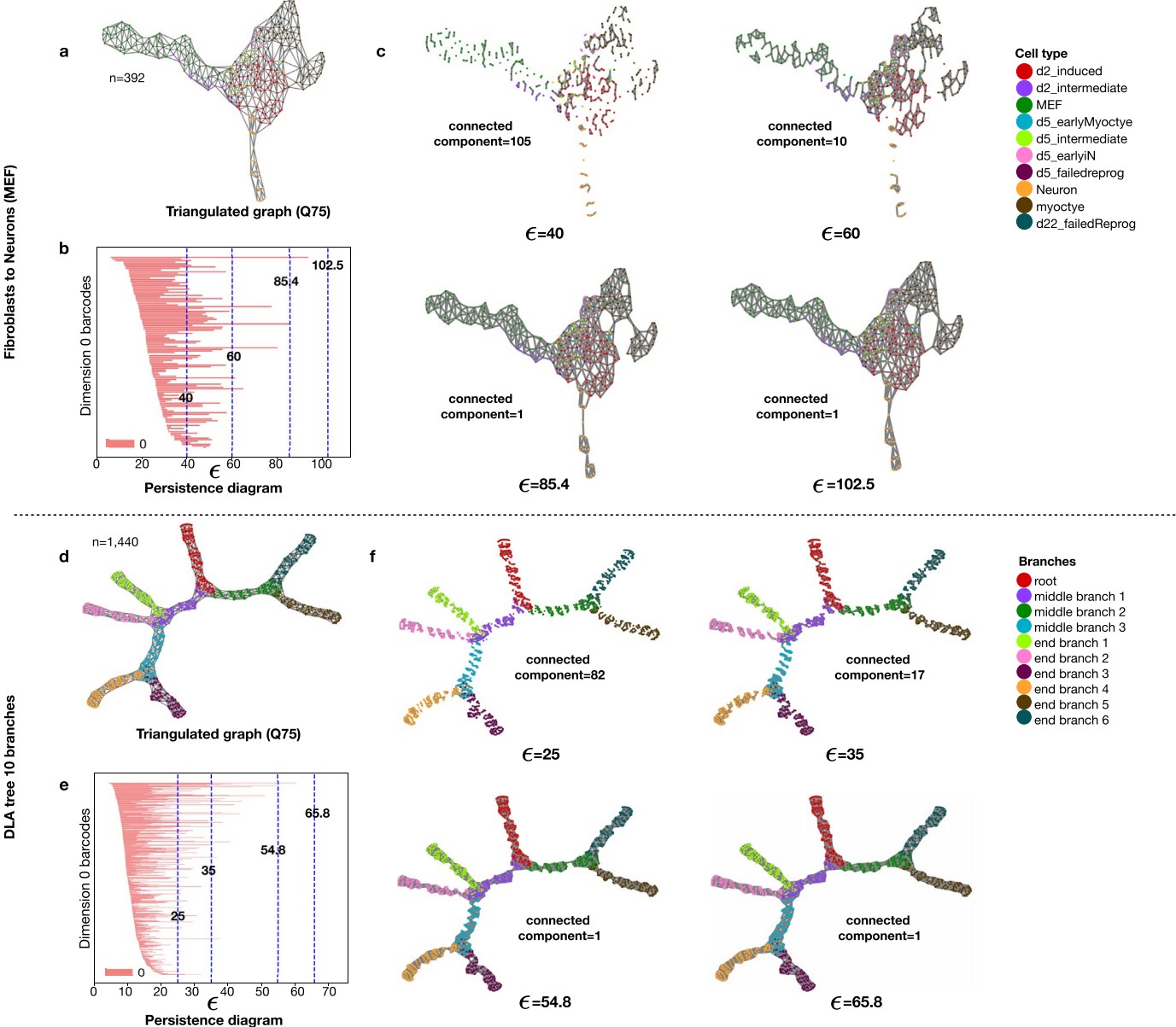

**Extended Data Fig. 4 | Persistent Homology Analysis. a**, We display the triangulated single cell graph (threshold Q75) for the MEF data. **b**, Persistence diagram with 0-loops for the same data cloud from (A). **c**, Graphs obtained after varying the the radius $\epsilon$ for several cutoffs values. We report the number of barcodes/connected components for each radius. The value 85.4 represents the first graph with a single connected component. Using an radius of factor 1.2 higher (102.5) leads to a graph with higher connectivity between nodes. This approach provides a graph similar to the filtering scheme used in PHLOWER (A). **d-f**, same as **a-c** for the DLA simulated data with 10 branches. The python module gudhi (3.10.1) is used for the persistence analysis.

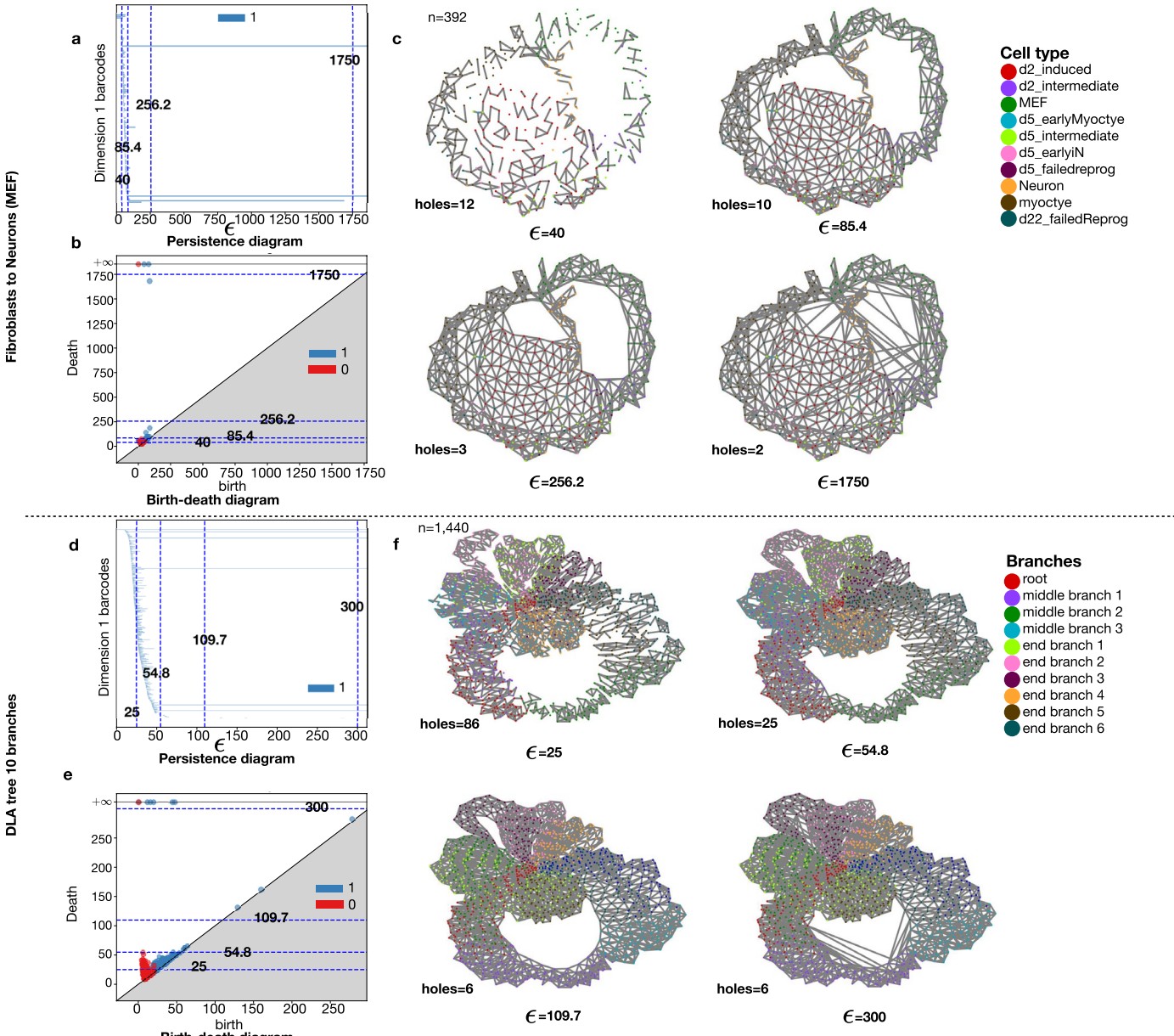

**Extended Data Fig. 5 | 1-dimensional Persistent Homology. a**, Persistence diagram with 1-loops estimated on the SC of the MEF cells. **b**, Birth and death diagram with both 0-loops (connected components) and 1-loops (holes) for the SC representing MEF cells. **c**, SCs obtained after varying the the radius $\epsilon$ for several cutoffs values. Interestingly, the radius of 300, which is the lowest to remove all 1-loops but those with "infinity radius" find two holes as expected in this data set. **d-f**, same as **a-c** for the simulated trees with 10 branches. In this more complex data set, the use of the threshold of 1750 provides a SC with 6 holes. This radius in the smallest radius such that only holes with "infinite size" are considered. These correspond to holes generated by artificially connecting cells with low and high pseudotime, precisely corresponding to features we want to extract with PHLOWER.

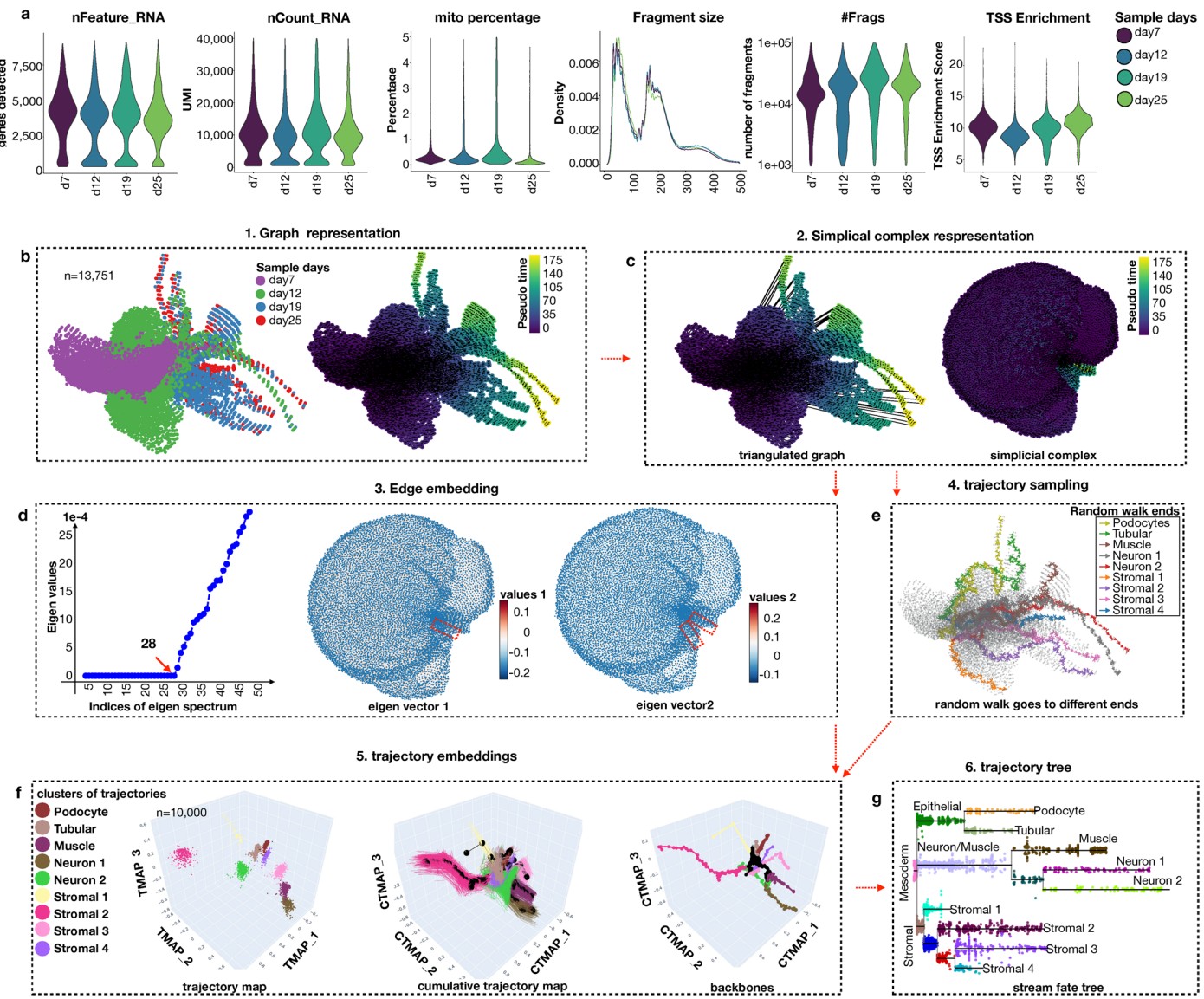

**Extended Data Fig. 6 | Quality check for kidney organoids single cell data.**
**a**, We show violin plots with quality check information after filtering and cell detection. These are: number of features (genes) in RNA, number of counts (transcripts) in RNA, proportion of mitochondrial genes (RNA), fragment sizes distribution (ATAC), number of fragments (ATAC) and transcription start site enrichment. All libraries had similar values across days of sampling.
**b-g**, PHLOWER workflow for the kidney organoids data as described in Extended Data Fig. 1.

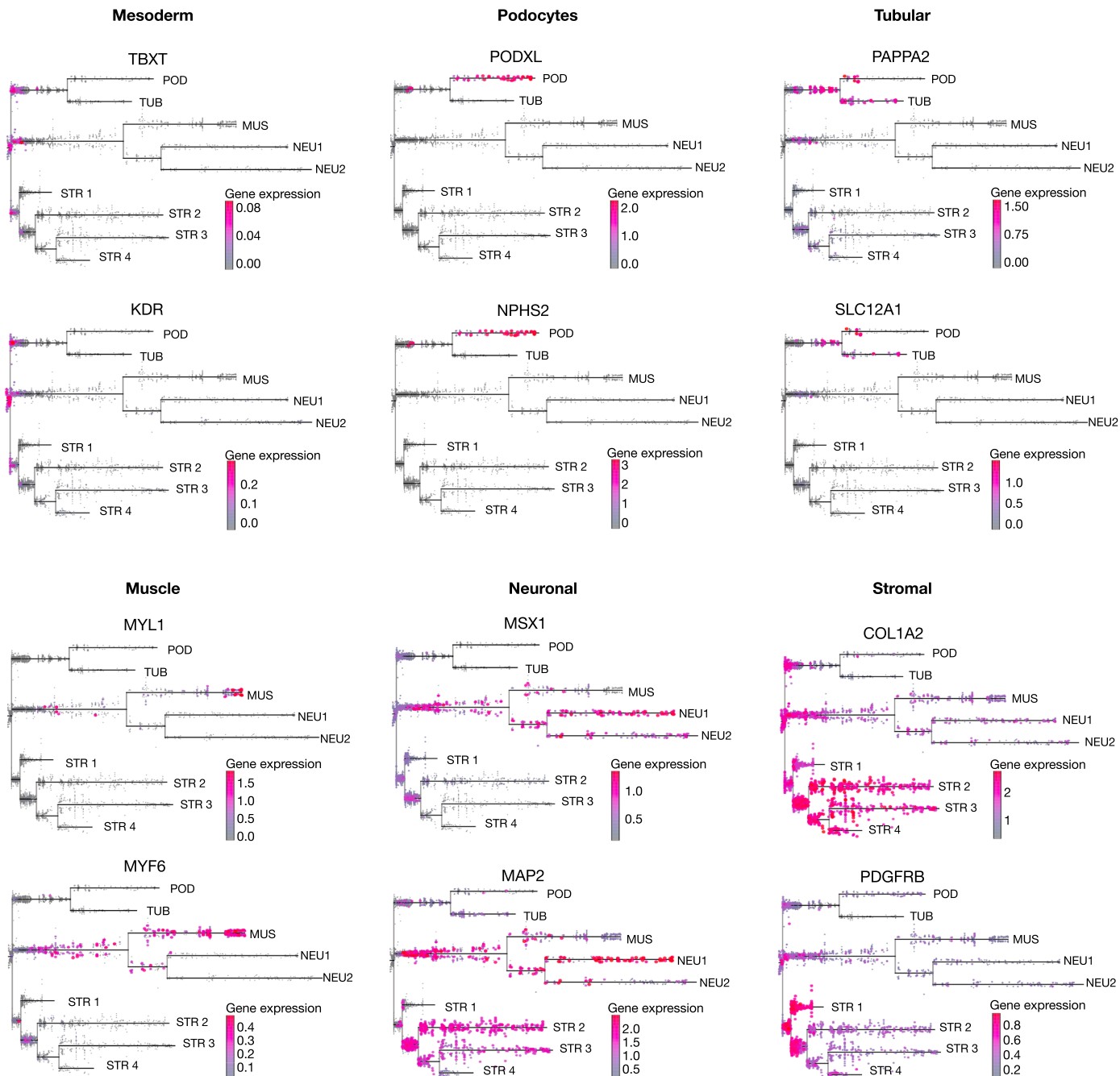

**Extended Data Fig. 7 | Gene markers for cell type of PHLOWER tree identification.** Expressions of the gene markers *TBXT, KDR, PODXL, NPHS2, PAPPA2, SLC12A1, MYL1, MYF6, MSX1, MAP2, COL1A2* and *PDGFRB* are shown in the PHLOWER tree (n=13,751; same genes as in Fig. 3d).

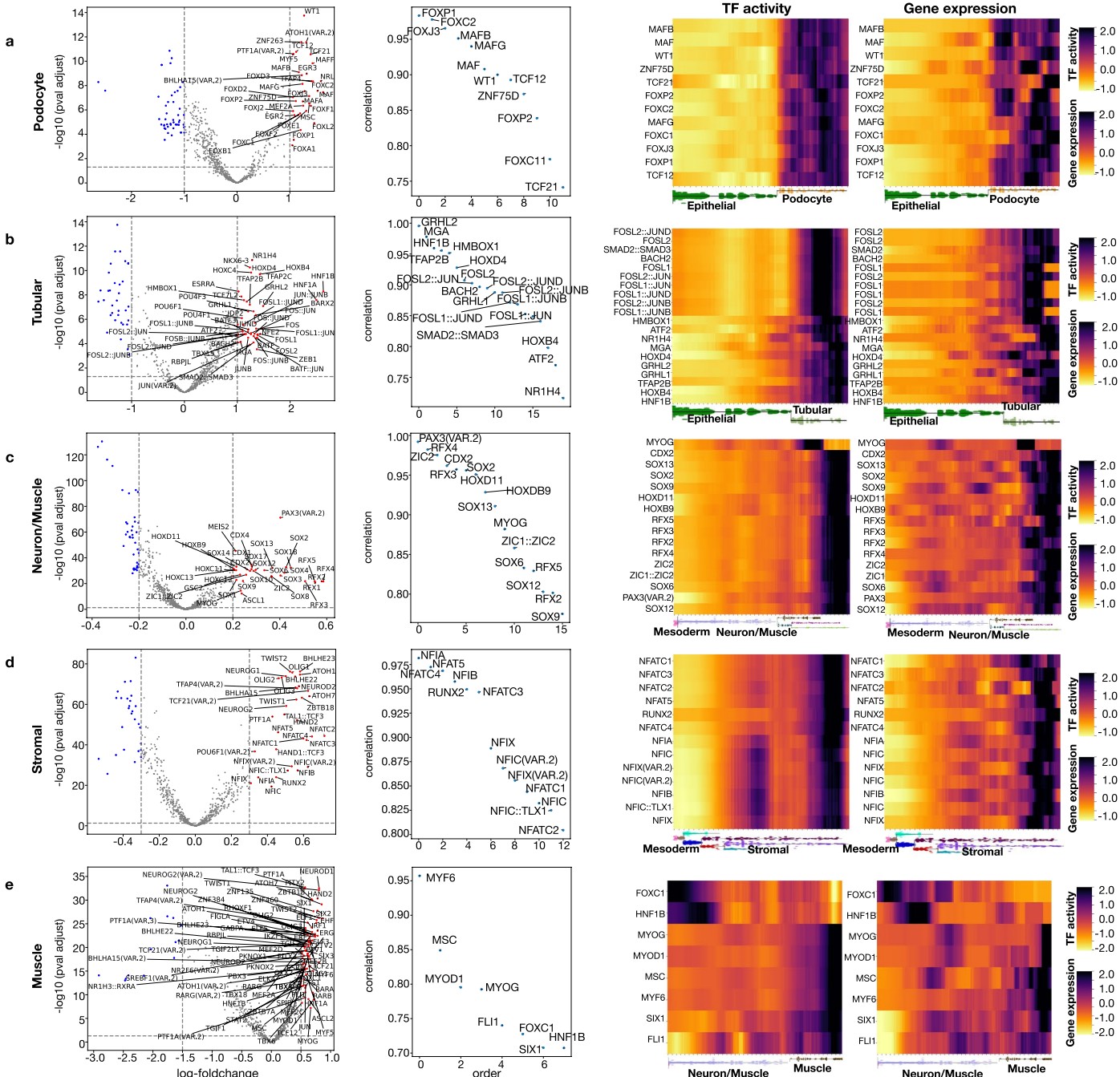

**Extended Data Fig. 8 | Regulators associated with cell branches. a**, Regulators associated with Podocyte cells. We perform a differential expression analysis to find genes specific to podocytes (comparing with tubular cells) (left). Of these DE genes, we select transcription factors, which transcription factor activity (scATAC-seq), is highly correlated with gene expression (middle). Heatmaps display the TF activity and gene expression profiles of these TFs over the differentiation path from mesoderm towards podocyte cells. **b**, Same as **a** when contrasting tubular cells with podocytes. **c**, Sames as **a** when contrasting neuronal/muscle cells with all other branches. **d**, Same as **a** when contrasting stromal cells with all other branches. **e**, Same as **a** when contrasting Muscle cells with all other Neuron cells.

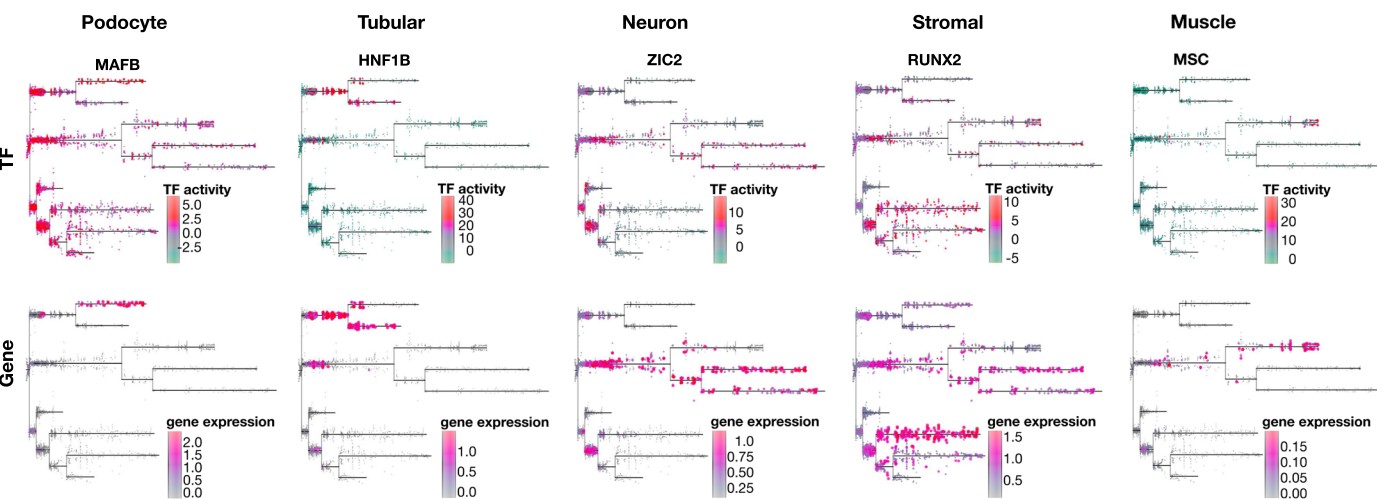

**Extended Data Fig. 9 | TF activity and gene expression of kidney organoid relevant transcription factors.** TF activity and gene expression of selected transcription factors in the PHLOWER estimated kidney organoid differentiation tree (n=13,751).

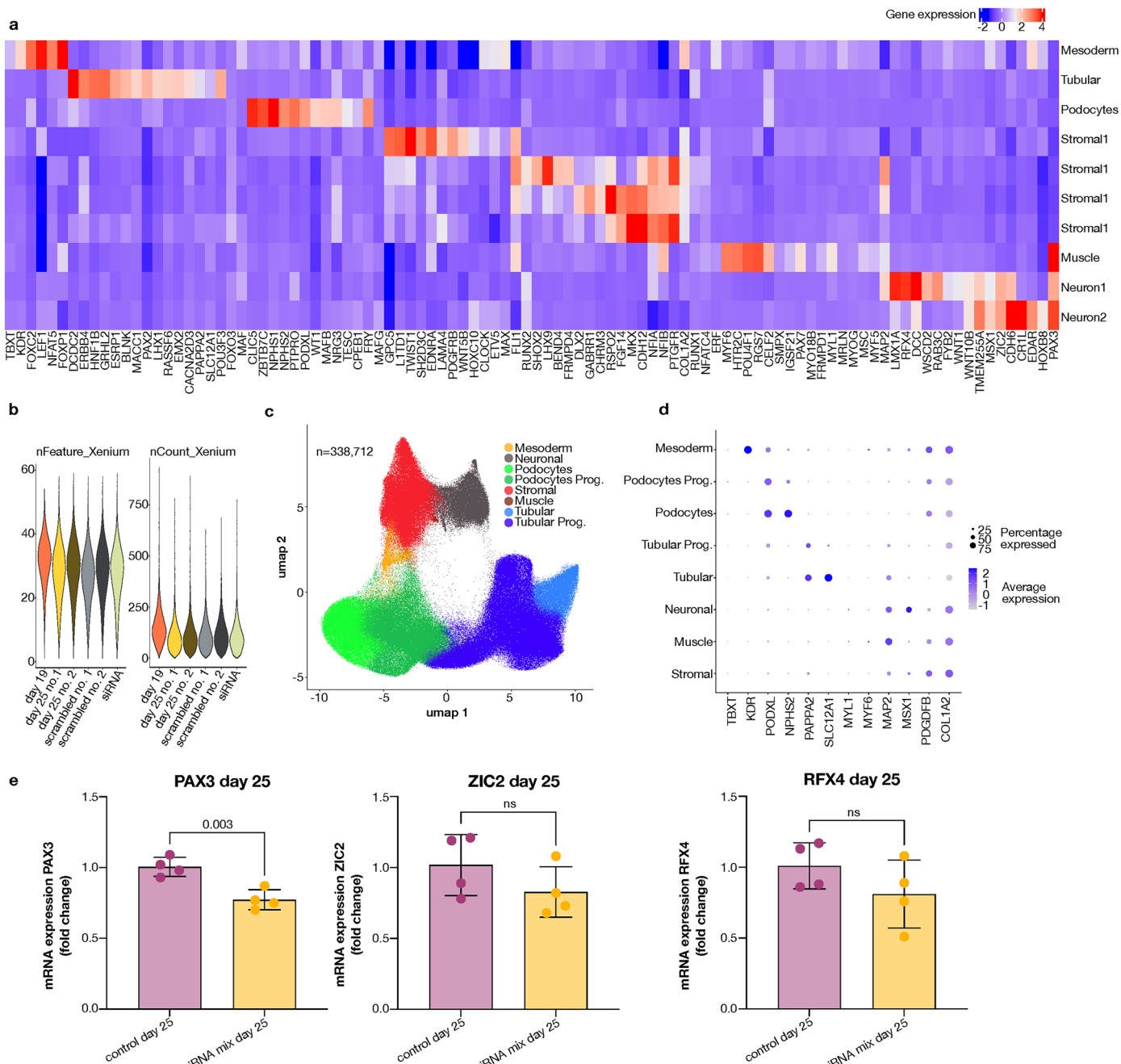

**Extended Data Fig. 10 | Xenium spatial profiling. a**, Gene expression in multiome kidney organoid data, highlighting the 100 genes and transcription factors selected for the Xenium experiment. **b**, QC for the day 19, day 25, scrambled siRNA and siRNA treated spatial data. **c**, UMAP shows the cell types of the xenium data. **d**, Dot plots show the markers of cell types of the xenium data. **e**, mRNA expression of genes targeted by siRNA experiments. Statistical test using unpaired two-tailed t-test with bars representing mean ± sd of at least N=3 per condition per experiment, 2 independent experiments.

# Reporting Summary

## Statistics

For all statistical analyses, confirm that the following items are present in the figure legend, table legend, main text, or Methods section.

| n/a | Confirmed | |
|-----|-----------|---|
| ☐ | ☒ | The exact sample size (*n*) for each experimental group/condition, given as a discrete number and unit of measurement |
| ☐ | ☒ | A statement on whether measurements were taken from distinct samples or whether the same sample was measured repeatedly |
| ☐ | ☒ | The statistical test(s) used AND whether they are one- or two-sided *Only common tests should be described solely by name; describe more complex techniques in the Methods section.* |
| ☐ | ☒ | A description of all covariates tested |
| ☐ | ☒ | A description of any assumptions or corrections, such as tests of normality and adjustment for multiple comparisons |
| ☐ | ☒ | A full description of the statistical parameters including central tendency (e.g. means) or other basic estimates (e.g. regression coefficient) AND variation (e.g. standard deviation) or associated estimates of uncertainty (e.g. confidence intervals) |
| ☐ | ☒ | For null hypothesis testing, the test statistic (e.g. *F*, *t*, *r*) with confidence intervals, effect sizes, degrees of freedom and *P* value noted *Give P values as exact values whenever suitable.* |
| ☒ | ☐ | For Bayesian analysis, information on the choice of priors and Markov chain Monte Carlo settings |
| ☒ | ☐ | For hierarchical and complex designs, identification of the appropriate level for tests and full reporting of outcomes |
| ☒ | ☐ | Estimates of effect sizes (e.g. Cohen's *d*, Pearson's *r*), indicating how they were calculated |

*Our web collection on statistics for biologists contains articles on many of the points above.*

## Software and code

Policy information about availability of computer code

| | |
|---|---|
| Data collection | no software was used for data collection. |
| Data analysis | We provide a Python package `PHLOWER`available at https://github.com/CostaLab/phlower/. All the trajectory inference and regulator discovery functions are implemented in `PHLOWER` package which can be installed by `git clone https://github.com/CostaLab/phlower/; cd phlower; pip install .`. We also built a website (https://phlower.readthedocs.io) where we provide the installation guide as well as tutorial of dataset `fib2neuron` and `kidney` we generated and presented in this manuscript. We also provide the benchmarking data as well as the multiome kidney organoid and xenium kidney organoid data presented in the manscript in zenodo:https://doi.org/10.5281/zenodo.13860460

Tools used in the preprocessing in the manuscript for multiome analysis:

R:
R v4.1.3
Seurat v3.2.3
ArchR v1.0.2
MOJITOO v1.0.0
Signac v1.9.0
chromVAR v1.20.0
DropletUtils v1.18.1
dplyr v1.1.2
ggplot2 v3.4.2 |

SingleCellExperiment 1.20.0

Tools used in the preprocessing in the manuscript for xenium analysis:
Seurat v5.0.2
ggplot2 v3.5.1
dplyr v1.1.4

Tools used in the data analysis in the manuscript for phlower:
Python:
Python v3.10.8
anndata v0.9.2
colorcet v3.0.1
matplotlib v3.9.1
networkx v2.8.8
numpy v1.23.5
pandas v2.2.3
pydot v1.4.2
scanpy v1.9.3
scipy v1.14.0
seaborn v0.13.2
sklearn v1.5.1

For manuscripts utilizing custom algorithms or software that are central to the research but not yet described in published literature, software must be made available to editors and reviewers. We strongly encourage code deposition in a community repository (e.g. GitHub). See the Nature Portfolio guidelines for submitting code & software for further information.

## Data

Policy information about availability of data

All manuscripts must include a data availability statement. This statement should provide the following information, where applicable:
- Accession codes, unique identifiers, or web links for publicly available datasets
- A description of any restrictions on data availability
- For clinical datasets or third party data, please ensure that the statement adheres to our policy

All pre-processed single cell and spatial data sets were deposited in zenodo (https://doi.org/10.5281/zenodo.13860460). Raw sequencing files have been deposited in GEO (GSE302266, GSE302264).

## Human research participants

Policy information about studies involving human research participants and Sex and Gender in Research.

| Reporting on sex and gender | N.A. |
| Population characteristics | N.A. |
| Recruitment | N.A. |
| Ethics oversight | N.A. |

Note that full information on the approval of the study protocol must also be provided in the manuscript.

# Field-specific reporting

Please select the one below that is the best fit for your research. If you are not sure, read the appropriate sections before making your selection.

☒ Life sciences    ☐ Behavioural & social sciences    ☐ Ecological, evolutionary & environmental sciences

For a reference copy of the document with all sections, see nature.com/documents/nr-reporting-summary-flat.pdf

# Life sciences study design

All studies must disclose on these points even when the disclosure is negative.

| Sample size | At least n=3 were used per organoid. No data point was excluded. |
| Data exclusions | N.A. |

| Replication | N.A. |
| Randomization | N.A. |
| Blinding | No blinding. |

# Reporting for specific materials, systems and methods

We require information from authors about some types of materials, experimental systems and methods used in many studies. Here, indicate whether each material, system or method listed is relevant to your study. If you are not sure if a list item applies to your research, read the appropriate section before selecting a response.

## Materials & experimental systems

| n/a | Involved in the study |
|-----|----------------------|
| ☐ | ☒ Antibodies |
| ☐ | ☒ Eukaryotic cell lines |
| ☒ | ☐ Palaeontology and archaeology |
| ☒ | ☐ Animals and other organisms |
| ☒ | ☐ Clinical data |
| ☒ | ☐ Dual use research of concern |

## Methods

| n/a | Involved in the study |
|-----|----------------------|
| ☒ | ☐ ChIP-seq |
| ☒ | ☐ Flow cytometry |
| ☒ | ☐ MRI-based neuroimaging |

## Antibodies

| Antibodies used | Primary antibodies: (NPHS1, catnr AF 4269-SP, RD Systems. Vimentin, catnr ab92547, Abcam. E-cadherin, catnr 610405, BD Biosciences. Secondary antibodies: donkey anti-sheep IgG (H+L) alexa fluor 647 (ThermoFisher), donkey anti-rabbit IgG (H+L) alexa fluor 488 (ThermoFisher), donkey anti-mouse IgG (H+L) alexa fluor 488 (ThermoFisher), DAPI (Sigma Aldrich, D9542) Merck). |
| Validation | The antibodies were validated in a previous study using human kidney tissue. Please refer to Jansen et al Cell Stem Cell 2022 doi: 10.1016/j.stem.2021.12.010, Jansen et al Develoment 2022 doi: 10.1242/dev.200198 and Takasato et al Nature 2015 doi.org/10.1038/nature15695. |

## Eukaryotic cell lines

Policy information about cell lines and Sex and Gender in Research

| Cell line source(s) | The iPS cells were reprogrammed from human somatic cells obtained from a healthy volunteer after given informed consent. |
| Authentication | no authentication was performed. |
| Mycoplasma contamination | The cells were tested negative for mycoplasma. |
| Commonly misidentified lines (See ICLAC register) | n.a. |

