## [Peer Review File · Nature Methods]

PHLOWER – Single-cell trajectory analysis using Hodge Decomposition

Corresponding Author: Professor Ivan Costa

A version of this paper was originally rejected for publication by Nature Methods, however that decision was reconsidered after appeal by the authors.

Version 0:

Decision Letter:

2nd Dec 2024

Dear Professor Costa,

Your Article entitled "PHLOWER - Single cell trajectory analysis using Decomposition of the Hodge Laplacian" has now been seen by 3 reviewers, whose comments are attached. While they find your work of potential interest, they have raised serious concerns which in our view are sufficiently important that they preclude publication of the work in Nature Methods, at least in its present form.

As you will see, the reviewers raise concerns about a lack of application to real world data as well as demonstrations of PHLOWER's broad utility and scalability.

Should further experimental data allow you to fully address these criticisms we would be willing to look at a revised manuscript (unless, of course, something similar has by then been accepted at Nature Methods or appeared elsewhere). This includes submission or publication of a portion of this work somewhere else. We hope you understand that until we have read the revised paper in its entirety we cannot promise that it will be sent back for peer-review.

If you are interested in revising this manuscript for submission to Nature Methods in the future, please contact me to discuss your appeal before making any revisions. Otherwise, we hope that you find the reviewers' comments helpful when preparing your paper for submission elsewhere.

Sincerely,
Madhura

Madhura Mukhopadhyay, PhD
Senior Editor
Nature Methods

Reviewers' Comments:

Reviewer #1:

Remarks to the Author:

In this manuscript, the authors develop PHLOWER model to analyze single cell trajectory. The key idea is to construct simplicial complexes based on cellular graph representation and then use Hodge Laplacian (HL) decomposition to infer trajectory embedding. The developed method is evaluated and benchmarked with state-of-the-art models. It has been found that PHLOWER model can outperform all existing models in both inference of complex trees and identification of transcription factors.

The model is novel, and the results are very convincing. I would recommend the publication of the paper if the following concerns were well addressed.

1) Differ from graph models, simplicial complexes are usually oriented. In particular, the construction of boundary matrix requires a rigorous definition of the orientation. Even though the authors mentioned statements like "edge e_j leaves vertex v_i " and "their directions follow the index of the vertices in increasing order". Is it still not clear what is the proper definition of the orientation? What is the proper definition of the vertex index? Note that even though the eigenvalues (of HL matrixes) do not rely on orientation, the eigenvectors are highly related to orientation.

2) For the construction of Delaunay triangulation, why only the 2D Delaunay triangulation is used? How about tetrahedron information? Are they useful?

Further, the authors "remove edges connecting distant cells". Why not just use VR complex or Alpha complex with a certain cutoff distance? Moreover, "75% quantile" is considered. Why choose to use 75% instead of other values? Is it possible to consider a series of HL matrixes as in persistent Laplacian (persistence spectral) models, such as

Zhenyu Meng and Kelin Xia, "Persistent spectral-based machine learning (PerSpect ML) for protein-ligand binding affinity prediction." *Science Advances*, 7 (19), eabc5329 (2021)

Rui Wang, Duc D Nguyen and Guo-Wei Wei, Persistent spectral graph, *International Journal for Numerical Methods in Biomedical Engineering*, 36(9), e3376 (2020).

3) The current model uses the harmonic generators (zero-value-related eigenvectors) from 1D HL matrix. Will it be helpful to consider harmonic generators from 2D HL matrix (or even higher-order ones)?

4) There are some mistakes and typos in the manuscripts, such as

a) Page 3, "cells with low (progenitors)"

b) Page 3, "the first two eigenvectors have zero eigenvectors"

c) Page 12, equations (10) and (11) are improper.

d) Page 12, " B_k which maps $k-1$ simplices to k simplices" should be from k to $k-1$.

e) Page 13, both equations (14) and (16) contain matrix D_1 , are they of the same meaning or slightly different?

The authors are suggested to go over their paper several times to remove potential mistakes.

Reviewer #2:

Remarks to the Author:

PHLOWER has problematic performance. In the comparison to other methods, the authors only used simulated data. This method basically adds dummy connections between the predicted starting and end points (from the previous step of pseudotime analysis) and then finds the 1-cycles. I would expect this to work well on the simulated data that almost perfectly resemble tree structures. The authors used metrics from Dynverse but did not use the real datasets from Dynverse (Dynverse does have a benchmark specifically for real and simulated data with tree structures). On the real dataset example (Figure 3A), the trajectory looks perfect because the cell types are assigned afterwards.

There is serious lack of innovation. The authors claim to fill the gap of considering multiomics. This was already considered in STREAM that they compared to. Also, nothing new was used for multimodal data. The authors simply used MOJITOO, another existing tool that gives a common embedding of multimodal data. The first part (multimodal sc data -> MOJITOO -> Knn graph -> Pseudotime) is very standard and the second part (hodge decomposition -> finding branches -> tree embedding) does not work well generally on real data with high noise, such as progenitor ductal cells from the dynverse benchmarking study and spatially resolved RNAseq sample (DLPFC sample which has a trajectory for neurogenesis and a branching between inhibitory and excitatory neuronal development).

There is no physical justification why the dimension 1 cycle from 1-Laplacian has to be associate with single cell branching dynamics. The whole procedure is artificial, the results are unphysical, and the approach fails for many real cases.

"The nonlinear manifold of gene expression space" was not defined.

"gene expression manifold" was not defined.

"The spectral decomposition of the HL can be used to decompose flows and trajectories 14, 16 on the edges into gradient, curl, and harmonic components of the gene expression manifold". This is not true for the graph Laplacian or combinatorial Laplacians used in this work defined on simplicial complex. Gene expression manifold was not defined anywhere in this work. Ref 14 does not show such a decomposition (I could find Ref 16). Only the decomposition of a vector field with REAL Hodge Laplacians defined on differentiable manifolds can give rise to DIVERGENT-FREE, CURL-FREE, and harmonic components (NOT "gradient, curl, and harmonic components").

Note that Hodge decomposition of single-cell RNA velocity was published, used REAL Hodge Laplacian defined on differentiable manifolds constructed from cell single-cell RNA velocity fields. One can see TRUE curl-free, divergence-free, and harmonic components of the RNA velocity field in a low dimensional representation. In contrast, I cannot find any of the claimed Hodge decomposition of "gradient, curl, and harmonic components of the gene expression manifold" in the present manuscript. The authors need to use precise mathematical concepts and terminologies to describe exactly what they are doing.

Note that graph Laplacian can be seen as a special case, i.e., the first one, of a series of combinatorial Laplacians introduced by Eckmann in 1944 (Harmonische funktionen und randwertaufgaben in einem komplex. *Commentarii Mathematici Helvetici*, 17(1):240-255). Hodge Laplacians and combinatorial Laplacians share the same homological algebraic structure but have totally different building blocks (simplicial complex on point clouds vs differential forms on manifolds).

In Fig. 1C, how can you color points in Eigenvector 1 and Eigenvector 2 with different eigen values? It appears to be either a low level mistake or misleading.

"The dominant eigenvectors of the zero-order Graph Laplacian are used to analyze this manifold". First, Graph Laplacian has no higher order ones and it is the zero-order one in combinatorial Laplacians. Additionally, the Laplacian is defined on simplicial complexes as the authors mentioned, rather than on a manifold as claimed.

In Fig. 1, does "edge embedding" mean a one-dimensional Laplacian-based embedding? Note that edges alone do not constitute the first-order combinatorial Laplacian because boundary operators connect them to lower- and upper-dimensional faces.

"Next, we perform a Hodge Decomposition of this simplex,..." How can the authors perform a Hodge decomposition on a simplex?

"... which provides edge level embeddings", What are edge level embeddings? As I mentioned, edges by themselves cannot define a topological dimension.

"as well as a trajectory embedding, where each embedding point represents a cell differentiation trajectory". What is the nature of the edge embedding? Is it an eigenvector, eigenvalue, or something else associated with the Hodge decomposition? If "edge embedding" refers to eigenvectors and associated eigenvalues as shown in the chart of Fig. 1C, then the term "edge embedding" is misleading from the mathematical point of view.

"The dominant eigenvectors of the associated graph Laplacian can then be used to analyse this gene expression manifold in more detail, in terms of clusters and latent dimensions."
What are "the dominant eigenvectors of the associated graph Laplacian"? What are the non-dominant eigenvectors of a graph Laplacian? What are latent dimensions?

In Figure S15 caption, the authors state that "We demonstrate the detection of 0-dimensional cycles (connected components) and 1-dimensional cycles (holes) using the 0-dimensional Hodge Laplacian and the 1-dimensional Hodge Laplacian decomposition". This statement is confusing. Why the detection of 0-dimensional cycles uses the 0-dimensional Hodge Laplacian, whereas the detection of 1-dimensional cycles uses the 1-dimensional Hodge Laplacian decomposition? What is the role of "decomposition" here? If the authors did the decomposition, what are the resulting coefficients of the projections?

It appears to me what the authors just solved (or diagonalized) the 0-combinatorial Laplacian matrix on the first input data (row 1, column 1) and the 1-combinatorial Laplacian matrix on the second input data (row 2, column 1) to obtain the harmonic eigenvalues and associated eigenvectors as well as non-harmonic eigenvalues and associated eigenvectors. This is not strictly a Hodge decomposition.

"whereas each of the top three 1-dimensional eigenvectors is around the three 1-dimensional cycles." This is confusing. What do the authors try to say here?

Ref. 56, "Modeling latent flows on single-cell data using the Hodge decomposition" was never published despite on the ArXiv for 5 years, which might be due to its conceptual problems.

In Fig. S17, how was the pseudo-time of the vertices (cells) defined?

PHLOWER was undefined in the title and abstract

The quality of many figures is very poor. I cannot clearly see many labels.

I do not know what the authors try to say: "In the MEF data, the first two eigenvectors have zero eigenvectors"

Remarks on code availability:

The code can run. But the tutorial is not very clear.

Reviewer #3:

Remarks to the Author:

The manuscript "PHLOWER - Single cell trajectory analysis using Decomposition of the Hodge Laplacian" revolves around the challenges of inferring complex, multi-branching cell differentiation trajectories from multi-modal single-cell sequencing data. A recent benchmarking study has revealed that current approaches can not address this task sufficiently. PHLOWER introduces a novel approach by leveraging simplicial complexes and Hodge Laplacian decomposition to create natural representations of cell differentiation processes, where branches manifest as holes in the manifold. The authors embed their tool together with another state-of-the-art tool STREAM into an existing benchmarking framework, where PHLOWER outperforms all competing methods on simulated datasets. The authors further apply PHLOWER to novel kidney organoid data for which scRNA-seq and scATAC-seq data are available and show that the branching recovered is in line with the

expectations and can identify transcription factors driving differentiation as well as off-target cells. The application of PHLOWER led to the identification of markers which were used in a gene panel for Xenium spatial transcriptomics at sub-cellular resolution. The PHLOWER-identified markers helped trace the spatial layout of kidney organoids. PHLOWER also identified some off-target related transcription factors and a knock-out experiment with siRNAs showed that these off-target cells (like neuronal cells) could be reduced to optimize the production of kidney organoids. PHLOWER makes an elegant use of a method that is not actually capable of inferring trees. While I'm not a mathematician, I could follow the explanations well. The manuscript is overall well written and understandable. The plots are mostly clear and underline the message of the paper well. The supplemental figures are also of high quality and support the manuscript further. The method has sufficient documentation and also the analysis is documented in a reproducible way. Data are shared through zenodo. The results of both the benchmark and the application to real-world data are convincing, although the latter is a showcase for which other tools might have revealed markers of similar quality. Nevertheless, I think this method will be a very valuable contribution for single-cell analysis. I have the following comments:

Major:

- It is nice to see that PHLOWER re-uses the dynverse benchmarking framework to evaluate their method, this is great. However, I feel that the evaluation falls behind what would have been possible. Specifically, the authors evaluated the methods on a relatively small simulation data set, neglecting the wealth of data sets used in the dynverse benchmark (i.e. gold or silver standard on 110 real and 229 synthetic datasets). Applying PHLOWER to a broad set of data sets would reveal strengths and limitations better and help to clarify if PHLOWER is a tool that should always be employed or if it is best used for complex trajectory inference problems only.
- The authors also did not compare their methods in terms of scalability with respect to the number of cells and features and the stability of the predictions after subsampling the datasets
- The authors did not comment on the runtime or memory requirements of their method compared to other methods.

Minor:

- This method has conceptual similarities to the commonly used UMAP method for dimensionality reduction. Since UMAPs are widely known in the single-cell domain, it may be worthwhile highlighting similarities and differences in the spectral embedding approach chosen here.
- I feel Figure 2A shows too many different colors and thus becomes a bit difficult to read. Maybe this plot can be simplified. Not sure it is worthwhile showing the #branches explicitly here. This could be done in a supplemental figure.
- The rationale part of the methods section is partially redundant with the introduction of the manuscript. This could be further streamlined.
- Page 4: "We leveraged here both the trees inferred by PHLOWER" - ...and STREAM - this sentence is incomplete.
- Page 5: "are differentially expressed" - are differentially expressed.
- Page 16: Typo in pseudo-time
- Please add software and package versions used.
- The links in the data availability section were broken.
- Adding a google colab for the tutorial would make it more accessible.

Remarks on code availability:

I have had a look at the code but did not run it. I think the authors should make an example available through google colab to make it easier to explore.

** For Nature Portfolio general information and news for authors, see <http://npg.nature.com/authors>.

Version 1:

Decision Letter:

Our ref: NMETH-A58008A-Z

3rd Jun 2025

Dear Ivan,

Thank you for submitting your response to our queries on your revised manuscript "PHLOWER – Single-cell trajectory analysis using Hodge Decomposition" (NMETH-A58008A-Z). It has now been seen by the original referees and their comments are below. The reviewers find that the paper has improved in revision, and therefore we'll be happy in principle to publish it in Nature Methods, pending minor revisions to satisfy the referees' final requests and to comply with our editorial and formatting guidelines. In the final version, please make sure to add these explanations to the paper.

We are now performing detailed checks on your paper and will send you a checklist detailing our editorial and formatting requirements within two weeks or so. Please do not upload the final materials and make any revisions until you receive this

additional information from us.

TRANSPARENT PEER REVIEW

ORCID

Sincerely,
Madhura

Madhura Mukhopadhyay, PhD
Senior Editor
Nature Methods

Reviewer #1 (Remarks to the Author):

All my questions have been well addressed. I have no further comments!

Reviewer #1 (Remarks on code availability):

NA

Reviewer #2 (Remarks to the Author):

The revised version may have improved a lot, but I still think it does not meet the standard of Nature Methods. Despite the use of mathematics, the method only moderately improves existing methods on a standard analysis task. My understanding of Nature Methods standard is there has to be at least one of the following (1) significant improvement on a standard task/utility or (2) providing new utility that can lead to new insights. This work does not anyone these standards.

1. There is confusion by the statements in the blue text. It was stated that PHLOWER is the top performer in all 8 combinations (two groups of datasets and four metrics).

It does not seem to outperform things on Fig. 2F and H. I believe these metrics on real datasets are more important than those on simulated data.

2. In the original Dynverse paper, PAGA and Slingshot significantly outperform all other methods (<https://www.nature.com/articles/s41587-019-0071-9/figures/2>). But Slingshot is no longer a high performer in this paper. Maybe this is because this paper uses a subset of read data in Dynverse with more complex structure.

Related to this, why not also include the performance of PHLOWER on the full Dynverse benchmark and use the evaluation values of other methods in the evaluation table in Dynverse paper? Are the authors deliberately hiding something?

3. Although improved, many mathematical statements are still not accurate.

Minor things:

1. I also identified typos after a quick glance of the blue text on benchmarking: (1) "PAGE" on top of page 6, (2) "20 dynverse tree inference algorithms" in caption of Figure 2. (3) "3.700 cells" and "18.000 cells" on bottom of page 4.
2. Many people may be fine with this, but I personally find it annoying to have many different capitalization of other methods

across the manuscript. For example, the capitalization of methods are respected in the black text on page 4 but not respected in the blue text and in Figure 2.

Reviewer #3 (Remarks to the Author):

The authors have done an admirable job at addressing all reviewer concerns, in particular as some reviewer comments were not very well supported.

Version 2:

Decision Letter:

17th Sep 2025

Dear Ivan,

I am pleased to inform you that your Article, "PHLOWER – Single-cell trajectory analysis using Hodge Decomposition", has now been accepted for publication in Nature Methods. The received and accepted dates will be 30th Sep 24 and 17th Sep 25. This note is intended to let you know what to expect from us over the next month or so, and to let you know where to address any further questions.

Over the next few weeks, your paper will be copyedited to ensure that it conforms to Nature Methods style. Once your paper is typeset, you will receive an email with a link to choose the appropriate publishing options for your paper and our Author Services team will be in touch regarding any additional information that may be required. It is extremely important that you let us know now whether you will be difficult to contact over the next month. If this is the case, we ask that you send us the contact information (email, phone and fax) of someone who will be able to check the proofs and deal with any last-minute problems.

Authors may need to take specific actions to achieve compliance with funder and institutional open access mandates.

If your research is supported by a funder that requires immediate open access (e.g. according to [Plan S principles](https://www.springernature.com/gp/open-science/plan-s-compliance) or the [NIH public access policy](https://www.springernature.com/gp/open-science/us-federal-agency-compliance)) then you should select the gold OA route, and we will direct you to the compliant route where possible. Because authors warrant under our subscription licensing terms that they haven't committed to licensing any version of their article under a licence inconsistent with the terms of our agreement – including the applicable embargo period – publication under the subscription model isn't suitable for authors whose funders require no embargo.

If you are active on Twitter/X or Bluesky, please e-mail me your and your coauthors' handles so that we may tag you when the paper is published.

Best regards,
Madhura

Madhura Mukhopadhyay, PhD
Senior Editor
Nature Methods

** Visit the Springer Nature Editorial and Publishing website at http://editorial-jobs.springernature.com?utm_source=ejP_NMeth_email&utm_medium=ejP_NMeth_email&utm_campaign=ejp_Nmeth > www.springernature.com/editorial-and-publishing-jobs for more information about our career opportunities. If you have any questions please click [here](mailto:editorial.publishing.jobs@springernature.com) .**

Open Access This Peer Review File is licensed under a Creative Commons Attribution 4.0 International License, which permits use, sharing, adaptation, distribution and reproduction in any medium or format, as long as you give appropriate credit to the original author(s) and the source, provide a link to the Creative Commons license, and indicate if changes were made. In cases where reviewers are anonymous, credit should be given to 'Anonymous Referee' and the source.

Dear Dr. Mukhopadhyay,

We thank you and the reviewers for your detailed comments on our manuscript, “*PHLOWER – Single-cell trajectory analysis using Hodge Decomposition.*” (N METH-A58008) We have extensively expanded the experiments and revised the manuscript to address all referees' concerns. Changes in the text are marked in blue.

In summary, we have:

- Expanded the benchmarking analysis by including 20 additional single-cell datasets. These new experiments further support PHLOWER's predictive power in inferring complex cell differentiation trees from single-cell data.
- Analyzed additional single-cell datasets, including neurogenesis and pancreas progenitor data commonly used in RNA velocity studies. In both cases, PHLOWER successfully recovers the known cell differentiation trees.
- Included a computational benchmarking analysis (time and memory requirements) and improved code accessibility by adding new tutorials and sharing Google Colab notebooks.
- Incorporated an analysis relating persistent homology and the Hodge Laplacian.
- Improved the notation and definitions of the methods.
- Clarified the fact that discrete Hodge Laplacians (defined on simplicial complexes) approximate continuous Hodge Laplacians (defined on vector fields);

Altogether, these changes strengthen the accuracy of PHLOWER in analyzing real single-cell data. Additionally, the revised manuscript provides a clearer contrast between the main methodological approaches explored by PHLOWER: the Hodge Laplacian defined on simplicial complexes, the Hodge Laplacian defined on vector fields, and persistent homology analysis. Finally, we show that PHLOWER represents a successful application of discrete Hodge Laplacians to a real-world problem.

We believe that the revised paper is a strong candidate for publication and will be of high interest to the readership of Nature Methods. Since the single-cell multimodal genomics and spatial transcriptomics fields are tremendously expanding across life sciences and cellular trajectories are one key delivery insight likely to attract many citations. Thank you for your consideration.

Sincerely, For all authors,

Ivan G. Costa & Rafal Kramann

Reviewers' Comments:

Reviewer #1:

Remarks to the Author:

In this manuscript, the authors develop PHLOWER model to analyze single cell trajectory. The key idea is to construct simplicial complexes based on cellular graph representation and then use Hodge Laplacian (HL) decomposition to infer trajectory embedding. The developed method is evaluated and benchmarked with state-of-the-art models. It has been found that PHLOWER model can outperform all existing models in both inference of complex trees and identification of transcription factors. The model is novel, and the results are very convincing. I would recommend the publication of the paper if the following concerns were well addressed.

1) Differ from graph models, simplicial complexes are usually oriented. In particular, the construction of boundary matrix requires a rigorous definition of the orientation. Even though the authors mentioned statements like “edge e_j leaves vertex v_i ” and “their directions follow the index of the vertices in increasing order”. Is it still not clear what is the proper definition of the orientation? What is the proper definition of the vertex index? Note that even though the eigenvalues (of HL matrixes) do not rely on orientation, the eigenvectors are highly related to orientation.

The reviewer is right that the chosen orientations may influence the eigenvectors associated to the simplicial complex. Note, however, that any choice of orientation will lead to consistent results as the chosen orientation is merely a matter of bookkeeping. Specifically, the edge (and triangle) orientations can be set arbitrarily via the following procedure: nodes are given numerical ids in order of creation; and edges and triangles are oriented towards nodes with higher id (see Supp. Fig. 22). When building our paths or “edge flow” (Eq. 22), we only consider paths between cells with an increase in pseudotime. We encode the signal as positive if the path and the initially chosen orientation match, and negative otherwise. If we were to choose another orientation of the edge initially, the sign of the entry would simply flip (in the eigenvector and the encoded signal) which leads to the same results. Essentially, choosing any orientation corresponds to choosing a particular “coordinate system” in which we measure flows. Within this coordinate system, all calculations are consistent and there is no need to choose a specific orientation. We have updated the section “4.5.2 Simplicial Complex” and the legend of Supp. Fig. 22 to make these aspects more clear.

2) For the construction of Delaunay triangulation, why only the 2D Delaunay triangulation is used?

How about tetrahedron information? Are they useful?

We have combined these two questions, as they are interlaced.

2D triangulation allows us to find triangles (2-simplices), while the 3D triangulation would allow us to characterize both triangles (2-simplices) and tetrahedra (3-simplices). Considering tetrahedrons would allow us to utilise 2-order Hodge Laplacians, which can reveal hulls of voids in the data. This is of clear importance in problems related to 3-dimensional data, i.e. modeling of molecule structures as explored in Meng et al. 2021. Trajectories, as explored in PHLOWER,

are however 1-dimensional by nature. The associated harmonic space only depends on simplices up to dimension 2 (triangles). The consideration of 3D triangulation would thus only impose unnecessary computational complexity to PHLOWER. This is of course an interesting topic for future research on novel deep profiling essays such as 3-dimensional spatial transcriptomics. We address this topic now in our discussion (Section 3).

Further, the authors “remove edges connecting distant cells”. Why not just use VR complex or Alpha complex with a certain cutoff distance? Moreover, “75% quantile” is considered. Why choose to use 75% instead of other values? Is it possible to consider a series of HL matrixes as in persistent Laplacian (persistence spectral) models, such as Zhenyu Meng and Kelin Xia, "Persistent spectral-based machine learning (PerSpect ML) for protein-ligand binding affinity prediction." *Science Advances*, 7 (19), eabc5329 (2021) Rui Wang, Duc D Nguyen and Guo-Wei Wei, Persistent spectral graph, *International Journal for Numerical Methods in Biomedical Engineering*, 36(9), e3376 (2020).

We now perform persistent homology analysis in our simplicial complexes. In short, we contrasted the “triangulated graph” (mentioned by the referee) with filtering using persistence analysis on the mouse embryonic fibroblasts (MEF) data and in a simulated data set (Supp. Fig. 19). We observe that the filtering step is equivalent to finding the “radius” to which the graph has only a single connected component. We further increase this radius by a ratio (20%) to increase the connectivity of edges, i.e. to avoid a sparse triangulation in the last edge connecting the SC.

Note that this procedure only requires the evaluation of 0-loops (connected components); as the “triangulated graph” is the step previous to connecting the initial and terminal cells to create holes and to generate the final simplicial complex. Of course, it is also interesting to evaluate if this procedure could be used to detect 1-loops (holes) in the final simplicial complex. We evaluate the same data sets as before in Supp. Fig. 20. We observe that in both MEF cell and simulated data the use of persistence and birth death diagrams detected the expected number of holes when a large enough radius lower than infinity is used (Supp. Fig. 20). This result is interesting, as it allows us to pick a large enough threshold such that only the ‘artificially created holes’ relevant to PHLOWER are alive. The birth-death diagram shows that there indeed is a range of thresholds that will guarantee this behaviour. Note however that persistent homology per se does not give unique generators that characterise all edges relevant for a hole and thus cannot replace the harmonic eigenvectors used in PHLOWER. This is of course an interesting topic of further research, which we refer to now in the discussion (Section 3).

3) The current model uses the harmonic generators (zero-value-related eigenvectors) from 1D HL matrix. Will it be helpful to consider harmonic generators from 2D HL matrix (or even higher-order ones)?

Hodge Laplacians of degree above 1 process data lying on faces, volumes, etc. As our method is concerned with trajectories which are defined on edges, 2D HLs will have no clear application here. (see also our answer to question 2 above) However, they have potential in other future applications such as 3 dimensional spatial transcriptomics and can be estimated similar to

1-HLs using our methods, although with higher computational cost. We have included this point in our discussion.

**4) There are some mistakes and typos in the manuscripts, such as
a) Page 3, “cells with low (progenitors)”**

We apologize and fixed this. It should read “low pseudotime”.

b) Page 3, “the first two eigenvectors have zero eigenvectors”

We fixed this. This should have been “zero eigenvalues”.

c) Page 12, equations (10) and (11) are improper.

Some terms of the equations had wrong superscripts. We have improved this part including the removal of an additional unnecessary step (former Eq. 11).

d) Page 12, “ B_k which maps $k-1$ simplices to k simplices” should be from k to $k-1$.

This has been fixed.

e) Page 13, both equations (14) and (16) contain matrix D_1 , are they of the same meaning or slightly different?

They have indeed distinct meanings and we improved the formulation to improve this. Please check new equations 13 and 15.

The authors are suggested to go over their paper several times to remove potential mistakes.

We have implemented all suggestions and carefully reviewed the manuscript. This included making the method definitions more consistent and clear.

Reviewer #2:

Remarks to the Author:

PHLOWER has problematic performance. In the comparison to other methods, the authors only used simulated data.

We respectfully disagree with the referee. Neither do we see a performance issue, nor do we use just simulated data. We showcase the predictive power of PHLOWER in a novel kidney organoid data; and used KO experiments to validate the function of predicted regulators

(transcription factors). These are presented in figures 3, 4 and 5 of our original submission. In our revision, we now also add 20 single cell RNA data provided by the benchmarking framework Dynverse, as well as analysis of two data sets with complex differentiation trees commonly used in RNA velocity analysis (see new Fig. 2 and corresponding text).

In short, we benchmark our method on simulated and real-world data of several kinds and our method performs well. Specifically, the benchmarking results indicate that our methods work for real-world and simulated data; PHLOWER was able to reconstruct cell differentiation trees with high accuracy, as well as to correctly allocate cells to branches. Altogether, these results clearly support the performance of PHLOWER in the inference of complex cell differentiation trees.

This method basically adds dummy connections between the predicted starting and end points (from the previous step of pseudotime analysis) and then finds the 1-cycles.

The detection of 1-cycles is only one step of PHLOWER (section 4.5). There are far more steps involved in the process of estimating cell differentiation trees from single cell data. This includes generation of cell graphs (section 4.3); pseudo-time estimation (sections 4.4); building a simplicial complex (section 4.5); Hodge Laplacian decomposition (also section 4.5); estimation of trajectory embeddings (section 4.6); and estimation of cell differentiation trees and allocation of the cell within the tree (also section 4.6).

I would expect this to work well on the simulated data that almost perfectly resemble tree structures.

First, our benchmarking shows that SOTA approaches still struggle with this apparently simple data. Fig. 2A, for example, indicates that the best three competing methods (Paga, Monocle and RaceID), which are widely used in the literature, only achieve a Hamming-Ipsen-Mikhailov distance of around 0.8 in the recovery of true structures.

In contrast, PHLOWER achieves a performance close to 1 on this synthetic data. However, PHLOWER is not limited to this “simple data”. As we show with added benchmarks of real scRNA-seq data, PHLOWER performs best in the identification of tree structures and allocation of cells to branches (Fig. 2E-H).

The authors used metrics from Dynverse but did not use the real datasets from Dynverse (Dynverse does have a benchmark specifically for real and simulated data with tree structures).

We initially did not consider Dynverse real-world single-cell data due to their simplicity, i.e. most data sets contained simple trees with very few branches and few cells per branch. This includes the MEF data set used as a simple example in Fig. 1. We have revisited the Dynverse repository and consider all unique data sets with at least 3 branches (see Supp. Table 1). These are now evaluated in Fig. 2E-H. Similarly as for the simulated data, PHLOWER performs best in all evaluated criteria. Regarding competing approaches, we also observe that some methods

perform well in simulated data sets but worse in real data sets (or vice-versa). Also, some methods do well on tree inference tasks, but not on cell allocation tasks. Altogether the results still support the performance of PHLOWER, which obtained best performance in all evaluated tasks.

We also evaluated PHLOWER in two data sets used in RNA velocity analysis (see answers below for more details).

On the real dataset example (Figure 3A), the trajectory looks perfect because the cell types are assigned Afterwards.

This analysis represents a real case study. We generated a novel data set; and used PHLOWER to find the trees and explore features (transcription factors and genes) explaining these. The branches of PHLOWER define the clusters, so they will per definition “look perfect”. The validation here is not based on the color of branches but on two aspects: does the tree reflect the expected biology; and can we find and test novel biological findings?

One example of known biology is the branch with epithelial cells, which bifurcate into tubular cells and podocytes as expected. We can further use PHLOWER to find features (transcriptions factor (TF) or genes) associated with these branches. As discussed in Fig. 3, PHLOWER indicates TFs WT1 and GRHL2 as main regulators of podocytes and tubular cells. These are both well known TFs related to these cells as supported by the biological literature.

Note that this example also shows the ability of PHLOWER to generate novel knowledge. PHLOWER predicts branches related to neuronal and muscle cells, which are considered as off-target cells and hamper kidney organoid development. We can use the prediction of PHLOWER to rank the TFs controlling the off-target cells, which indicated TFs PAX3, RFX4 and ZIC2. This information is a novel finding derived from PHLOWER. We performed wet lab experiments to show that by perturbing the expression of these TFs improves organoid development . These validation experiments, which require extensive lab work, are presented in Fig. 5.

There is a serious lack of innovation.

We strongly disagree with this opinion from the referee. The main novel aspect of PHLOWER is to explore the harmonic component of Hodge Laplacians to estimate trajectory embeddings and infer cell differentiation trees. We are not aware of any computational approach able to estimate edge or path level embeddings on single cell data. Our benchmarking and validations demonstrate the usefulness of this novel approach compared to state-of-the art approaches.

The authors claim to fill the gap of considering multiomics. This was already considered in STREAM that they compared to.

We are not aware of any version of stream (<https://github.com/pinelloolab/STREAM>) supporting multimodal data.

Also, nothing new was used for multimodal data. The authors simply used MOJITOO, another existing tool that gives a common embedding of multimodal data. The first part (multimodal sc data -> MOJITOO -> Knn graph -> Pseudotime) is very standard.

These initial steps from PHLOWER build upon state-of-the-art approaches for trajectory analysis in single-cell data as is completely common in many of the trajectory inference tools found in the literature and evaluated here. All these steps are crucial for the estimation of the graph (Sections 4.3 and 4.4); and we refer to original works introducing these in our manuscript. We do not claim these to be novel methods. Building upon previous works is standard practice in our field. Note further that MOJITOO is an approach proposed by us to estimate joint-embedding of multimodal single cell data. This was recently indicated to be SOTA in a review (doi.org/10.5281/zenodo.10540843). Also, none of the competing approaches evaluated in Fig. 2 is able to work with multimodal data.

and the second part (hodge decomposition -> finding branches -> tree embedding) does not work well generally on real data with high noise, such as progenitor ductal cells from the dynverse benchmarking study and spatially resolved RNAseq sample (DLPFC sample which has a trajectory for neurogenesis and a branching between inhibitory and excitatory neuronal development).

The main novel aspect of PHLOWER is the construction of the simplicial complex from single cell data and estimation trajectory embeddings (Sections 4.5 and 4.6). We were surprised by this statement of the referee that PHLOWER “*does not work well generally on real data with high noise*”, which is not substantiated or further explained by the referee.

Our original submission included the analysis of multiome kidney organoid data (see point above). We are not aware of any evaluation of PHLOWER in either a progenitor ductal cell or a “DLPFC” spatial data, as suggested by the referee. We are also not aware of any progenitor ductal cell data in Dynverse (see the list of Dynverse data here <https://zenodo.org/records/1443566>). We assume the referee refers to the pancreas progenitor and neurogenesis data evaluated and analysed in CellRank (doi.org/10.1038/s41592-021-01346-6) and scVelo (<https://doi.org/10.1038/s41587-020-0591-3>).

To further support the use in single cell data, we now include an evaluation of real scRNA-seq data provided by the Dynverse platform to further substantiate that the referee is indeed not correct here (see Fig. 2E-H and discussion above). We include an analysis of PHLOWER (and competing approaches) on pancreas progenitor data and the neurogenesis data. Since these data sets are not part of Dynverse, we cannot perform an in-depth benchmarking analysis as shown in Fig.2A-H. We now include the inferred tree from PHLOWER and competing approaches in Fig. 2I-J and Supp. Fig. 6-7. We can observe that PHLOWER recovers the expected branching trees in both data sets, while competing methods as monocle3 found

unconnected trees; and PAGA predicted non-existing branches in the neurogenesis data sets. Altogether, we have expanded the evaluation of real single-cell data on 22 additional data sets, which all support the value of PHLOWER in the inference of complex tree trajectories.

There is no physical justification why the dimension 1 cycle from 1-Laplacian has to be associated with single cell branching dynamics. The whole procedure is artificial, the results are unphysical, and the approach fails for many real cases.

First, we do not assume any physical interpretation of the 1-cycle and there does not need to be a physical interpretation as well. The high-dimensional vector space of expression data we consider is of course a mathematical abstraction, there is nothing “physical” about this space and neither does there have to be something physical. The fact that the null space of the 1-Laplacian can be associated to harmonic flows on edges is a well-established mathematical fact.

Second, we are not aware of any evaluation of PHLOWER supporting the statement of the referee that “the approach fails in many real cases.” -- this assertion by the referee has no support and in fact, we demonstrate more evidence in the revised manuscript that PHLOWER performs well in many real cases.

“The nonlinear manifold of gene expression space” was not defined. “gene expression manifold” was not defined. “The spectral decomposition of the HL can be used to decompose flows and trajectories^{14, 16} on the edges into gradient, curl, and harmonic components of the gene expression manifold”. This is not true for the graph Laplacian or combinatorial Laplacians used in this work defined on simplicial complex. Gene expression manifold was not defined anywhere in this work. Ref 14 does not show such a decomposition (I could find Ref 16). Only the decomposition of a vector field with REAL Hodge Laplacians defined on differentiable manifolds can give rise to DIVERGENT-FREE, CURL-FREE, and harmonic components (NOT “gradient, curl, and harmonic components”).

We believe that most of these terms are well-understood by readers within our field of study. Indeed, the idea of a nonlinear manifold underlying gene expression goes back to Waddington and has been discussed in many studies in this area -- of which we believe almost none defines mathematically precisely what a manifold actually is. Ref. [14] and indeed many other works considering discrete Hodge-Laplacians provide a clear account of how the space of edge-signals can be decomposed into (appropriately defined) gradient-free, curl-free and harmonic components that can be intuitively interpreted.

Nonetheless, we have overworked the manuscript to improve and clarify definitions. Among others, we restrained from using the term “gene expression manifold”, we clarified that PHLOWER focuses on the harmonic component of the Hodge Decompositions, and we improved some wrong/misleading terms such as “zero-order Graph Laplacians”. We have

overworked sections 1, 2.1 and 4 to reflect this. See comments below regarding further comments.

Note that Hodge decomposition of single-cell RNA velocity was published, used REAL Hodge Laplacian defined on differentiable manifolds constructed from cell single-cell RNA velocity fields. One can see TRUE curl-free, divergence-free, and harmonic components of the RNA velocity field in a low dimensional representation. In contrast, I cannot find any of the claimed Hodge decomposition of “gradient, curl, and harmonic components of the gene expression manifold” in the present manuscript. The authors need to use precise mathematical concepts and terminologies to describe exactly what they are doing.

We also understand that a recurrent critique of this reviewer concerns the distinction between Hodge Laplacians defined on vector fields (or differential forms on manifolds) vs. the “discrete counterpart” Hodge Laplacians defined on simplicial complexes. The latter is also named combinatorial Laplacian by some in the field, while there is a vast literature adopting our definitions of “discrete Hodge Laplacian”, e.g. doi.org/10.1137/18M1223101, doi.org/10.1137/18M1201019, doi.org/10.1063/5.0080370 just to cite a few examples.

There are guarantees that the behaviour of the Discrete Hodge Laplacian, when weighted accordingly, converges to the continuous Hodge Laplacian in the limit, for example (Dudziuk, “Finite-difference approach to the Hodge theory of harmonic forms”, 1976 doi.org/10.2307/2373615). Latter, Chen et al., 2023 (doi.org/10.48550/arXiv.2103.07626) give then guarantees on the estimation of the 1-Laplacian on Simplicial Complexes from points sampled from a manifold. Ribando-Gros and colleagues (<https://doi.org/10.1137/22M1482299>) show that unweighted Hodge Laplacians on simplicial complexes built on point clouds sampled from a point cloud do not necessarily approximate the spectrum of the continuous Hodge Laplacians on differential forms on the underlying manifold. Note that in our case, the point cloud is taken from an embedding where the metric does not have an intrinsic meaning. We are only interested in our Normalised Hodge Laplacian with associated Hodge Theory in the constructed simplicial complex. Our harmonic eigenvectors are not constructed to approximate some physical ground truth (which is not known in the context of single cell sequencing anyhow), but are used to generate trajectory embeddings, which are later used to estimate differentiation trees. This choice is validated by the strong empirical performance of PHLOWER on both simulated and real data sets.

We have now adapted our text to reflect the fact that we focus on the harmonic component of the discrete Hodge Laplacian (see below). We have decided to keep our current nomenclature, as this is already adopted and accepted by most of our peers. We refer in the discussion (Section 3) about the distinct types of Hodge Laplacians.

The reviewer did not specify which works explore Hodge Laplacians and single cell RNA velocity. While Chen et al., 2023 (doi.org/10.48550/arXiv.2103.07626) explores RNA velocity with the use of discrete Hodge Laplacians, we assume that the reviewer referees to the work

(Su et al. 2024; [10.1021/acs.jcim.4c00132](https://doi.org/10.1021/acs.jcim.4c00132)). There, authors explore properties of the Hodge Laplacian decomposition of RNA velocity fields (projected into UMAP and t-SNE) into gradient-free, curl-free and harmonic components. Note that this work focuses on visualization aspects of RNA Velocity fields, and is not used to infer underlying differentiation trees or allocate cells into trees as PHLOWER. Therefore, it is not directly comparable to PHLOWER. Interestingly, the authors state in their conclusion *“By the Hodge decomposition, one can extract these dynamic features in the resulting decomposed components of the original velocity field, for example, the circular shapes in its divergent-free component that are associated with the cell cycle process and the overall flow direction in the harmonic component that provides the cell lineage information.”* This is in line with PHLOWER, which also explores the harmonic components of the discrete Hodge Laplacian to delineate cell lineages as performed by PHLOWER. We refer to this interesting approach in our discussion (Section 3).

Note that graph Laplacian can be seen as a special case, i.e., the first one, of a series of combinatorial Laplacians introduced by Eckmann in 1944 (Harmonische funktionen und randwertaufgaben in einem komplex. Commentarii Mathematici Helvetici, 17(1):240–255). Hodge Laplacians and combinatorial Laplacians share the same homological algebraic structure but have totally different building blocks (simplicial complex on point clouds vs differential forms on manifolds).

See the previous point.

In Fig. 1C, how can you color points in Eigenvector 1 and Eigenvector 2 with different eigen values? It appears to be either a low level mistake or misleading.

There was an error in the legend, which should read “eigenvectors” instead of “eigen values”.

“The dominant eigenvectors of the zero-order Graph Laplacian are used to analyze this manifold”. First, Graph Laplacian has no higher order ones and it is the zero-order one in combinatorial Laplacians. Additionally, the Laplacian is defined on simplicial complexes as the authors mentioned, rather than on a manifold as claimed.

We revised the sometimes not fully consistent nomenclature when referring to Graph and Hodge Laplacians. We have reviewed the complete text regarding this.

In Fig. 1, does “edge embedding” mean a one-dimensional Laplacian-based embedding? Note that edges along do not constitute the first-order combinatorial Laplacian because boundary operators connect them to lower- and upper-dimensional faces.

The edge embedding refers to the harmonic components, i.e. eigenvectors with zero eigenvalues of the 1st order Hodge Laplacian of SCs. We have re-organized the text to make this more explicit. The improved definitions are found in Section 4.5.3 and Eq. 19.

“Next, we perform a Hodge Decomposition of this simplex,...” How can the authors perform a Hodge decomposition on a simplex?

This should read “Next, we perform a Hodge Decomposition of this simplicial complex”.

“... which provides edge level embeddings”, What are edge level embeddings? As I mentioned, edges by themselves cannot define a topological dimension.

The edge embedding refers to the harmonic eigenvector defined in Eq. 19. We have improved the text to explicitly state equations related to distinct components of PHLOWER.

“as well as a trajectory embedding, where each embedding point represents a cell differentiation trajectory”. What is the nature of the edge embedding? Is it an eigenvector, eigenvalue, or something else associated with the Hodge decomposition? If “edge embedding” refers to eigenvectors and associated eigenvalues as shown in the chart of Fig. 1C, then the term “edge embedding” is misleading from the mathematical point of view.

The trajectory embedding is defined by Eq. 21, which is obtained by the projection of the edge flows (Eq. 20) into the harmonic eigenvectors (Eq. 19). We have reorganized the text around Sections 4.5.3 and 4.6 for improving clarity. Fig. 1C and Supp. Figs. 1C, 2C, 9C show the harmonic eigenvectors (eigenvectors with zero eigenvalues) in the simplicial complex. Examples of the trajectory or edge-flow embedding, where every dot represents an edge flow, are shown in S1E, S2E and S8E.

“The dominant eigenvectors of the associated graph Laplacian can then be used to analyse this gene expression manifold in more detail, in terms of clusters and latent dimensions.” What are “the dominant eigenvectors of the associated graph Laplacian”? What are the non-dominant eigenvectors of a graph Laplacian? What are latent dimensions?

We referred to eigenvectors with zero eigenvalues of the graph Laplacian. We have rephrased the corresponding text for clarity.

In Figure S15 caption, the authors state that “We demonstrate the detection of 0-dimensional cycles (connected components) and 1-dimensional cycles (holes) using the 0-dimensional Hodge Laplacian and the 1-dimensional Hodge Laplacian decomposition”. This statement is confusing. Why the detection of 0-dimensional cycles uses the 0-dimensional Hodge Laplacian, whereas the detection of 1-dimensional cycles uses the 1-dimensional Hodge Laplacian decomposition? What is the role of “decomposition” here? If the authors did the decomposition, what are the resulting coefficients of the projections? It appears to me what the authors just solved (or diagonalized) the 0-combinatorial Laplacian matrix on the first input data (row 1, column 1) and the 1-combinatorial Laplacian matrix on the second input data (row 2, column 1) to

obtain the harmonic eigenvalues and associated eigenvectors as well as non-harmonic eigenvalues and associated eigenvectors. This is not strictly a Hodge decomposition. ``whereas each of the top three 1-dimensional eigenvectors is around the three 1-dimensional cycles." This is confusing. What do the authors try to say here?

The idea of this figure is to contrast the eigenvectors of Graph Laplacian and the harmonic eigenvectors of the Hodge Laplacian. We have improved the legend of the new Supp. Fig. 21, as well as text referring to this figure to improve this.

Ref. 56, ``Modeling latent flows on single-cell data using the Hodge decomposition" was never published despite on the ArXiv for 5 years, which might be due to its conceptual problems.

We are not involved in the above mentioned study, and we are not aware of the reasons it has not been published beyond ArXiv. Note that the refereed work focuses on the divergence component of the Hodge Laplacian, which can give smoothed estimates of the pseudo-time. The work does not explore the harmonic component; or does not infer differentiation trees. Moreover, it does not provide any benchmarking and only performs anecdotal analysis of simple data sets such as the MEF differentiation used by us in Fig. 1.

In Fig. S17, how was the pseudo-time of the vertices (cells) defined?

Pseudo-time of cells is estimated as indicated in Eq. 10.

PHLOWER was undefined in the title and abstract

We have defined it in the abstract.

The quality of many figures is very poor. I cannot clearly see many labels.

We have increased the size and changed the figure organization (in particular Fig. 2) to improve readability.

I do not know what the authors try to say: ``In the MEF data, the first two eigenvectors have zero eigenvalues"

These should read "the first two eigenvectors of the harmonic component have zero eigenvalues".

Remarks on code availability: The code can run. But the tutorial is not very clear.

We have improved the tutorial, which is expanded to five data sets explored in the main manuscript: MEF, kidney, pancreas progenitor, neurogenesis and a simulated data set. These are visible at:

<https://phlower.readthedocs.io/en/latest/tutorial.html>

Reviewer #3:

Remarks to the Author:

The manuscript "PHLOWER - Single cell trajectory analysis using Decomposition of the Hodge Laplacian" revolves around the challenges of inferring complex, multi-branching cell differentiation trajectories from multi-modal single-cell sequencing data. A recent benchmarking study has revealed that current approaches can not address this task sufficiently. PHLOWER introduces a novel approach by leveraging simplicial complexes and Hodge Laplacian decomposition to create natural representations of cell differentiation processes, where branches manifest as holes in the manifold. The authors embed their tool together with another state-of-the-art tool STREAM into an existing benchmarking framework, where PHLOWER outperforms all competing methods on simulated datasets. The authors further apply PHLOWER to novel kidney organoid data for which scRNA-seq and scATAC-seq data are available and show that the branching recovered is in line with the expectations and can identify transcription factors driving differentiation as well as off-target cells. The application of PHLOWER led to the identification of markers which were used in a gene panel for Xenium spatial transcriptomics at sub-cellular resolution. The PHLOWER-identified markers helped trace the spatial layout of kidney organoids. PHLOWER also identified some off-target related transcription factors and a knock-out experiment with siRNA showed that these off-target cells (like neuronal cells) could be reduced to optimize the production of kidney organoids. PHLOWER makes an elegant use of a method that is not actually capable of inferring trees. While I'm not a mathematician, I could follow the explanations well. The manuscript is overall well written and understandable. The plots are mostly clear and underline the message of the paper well. The supplemental figures are also of high quality and support the manuscript further. The method has sufficient documentation and

also the analysis is documented in a reproducible way. Data are shared through zenodo. The results of both the benchmark and the application to real-world data are convincing, although the latter is a showcase for which other tools might have revealed markers of similar quality. Nevertheless, I think this method will be a very valuable contribution for single-cell analysis. I have the following comments:

Major:

- It is nice to see that PHLOWER re-uses the dynverse benchmarking framework to evaluate their method, this is great. However, I feel that the evaluation falls behind what would have been possible. Specifically, the authors evaluated the methods on a relatively small simulation data set, neglecting the wealth of data sets used in the dynverse benchmark (i.e. gold or silver standard on 110 real and 229 synthetic datasets). Applying PHLOWER to a broad set of data sets would reveal strengths and limitations better and

help to clarify if PHLOWER is a tool that should always be employed or if it is best used for complex trajectory inference problems only.

The dynverse study, which dates back to 2018, has sparse simulated and real data sets with few cells and simple branchings. We have now revisited the real data sets from Dynverse; and we include in our benchmarking 20 real data sets (Supp. Table 1). These data have at least 3 branches and a data set is only included once, i.e. we do not include data sets based on sub-sampling of larger data sets to avoid bias. The performance of PHLOWER in this benchmarking study is similar to the simulated data (Fig. 2E-H). On the other hand, we observe that none of the competing methods performs well on the distinct evaluated tasks: some performed better in simulated data or in real data; while some have a better performance in the tree inference problems vs. cell allocation problem.

We also include an analysis of two data sets recurrently used in RNA velocity studies: a pancreas progenitor and neurogenesis data sets (Fig. 2I-J; Supp. Fig. 6 and 7), which are not part of dynverse. Therefore, we cannot evaluate the results using the statistics as presented in Fig. 2A-H. Results show that PHLOWER can detect trees recapitulating the expected trees, while competing approaches fail in most cases (see new Fig. 2 and corresponding text). Altogether, these new results, together with our previous analysis and validation in kidney organoids, provide strong evidence of the power of PHLOWER in the inferred of trees with at least three branches.

- The authors also did not compare their methods in terms of scalability with respect to the number of cells and features and the stability of the predictions after subsampling the datasets

- The authors did not comment on the runtime or memory requirements of their method compared to other methods

We answer these two questions together, as they are closely related. We now include a time and memory use profiling of all benchmarked data sets in the pancreas progenitor and neurogenesis data sets. Note that both the number of cells and the size/complexity of the three are the two major players in the run time of algorithms. This is shown in Supp. Fig. 4. PHLOWER needs 0.5 hour for a data set with 3.7k cells, and 12 hours for a data set with 18k cells. The memory requirements were 23 GB and 34 GB respectively. Experiments were performed in a standard desktop computer.

Next, we applied a sub-sampling strategy to the Neurogenesis data, starting with 18k cells and reducing it to 6k cells (Supp. Fig. 5). This resulted in an 8.6x speedup and the use of only 1/6 of the memory. Note that these experiments were conducted on an HPC node with more cores per CPU, which provided a 3x speedup compared to the performance on a desktop computer. However, the use of the HPC was not possible for the Dynverse experiments due to technical constraints, as it required superuser permissions. We also present PHLOWER stream trees estimated on these distinct datasets, which recover similar structures across the different

sub-sampled sets. Altogether, these results show that PHLOWER provides accurate predictions even when considering only a third of the cells.

Minor:

- This method has conceptual similarities to the commonly used UMAP method for dimensionality reduction. Since UMAPs are widely known in the single-cell domain, it may be worthwhile highlighting similarities and differences in the spectral embedding approach chosen here.

While UMAPs also explore topological properties of single cell graphs, they work on distinct dimensions. PHLOWER embeddings are defined on trajectories, i.e. one point represents a trajectory, while UMAPs embeddings are defined in the cells. So, they are not comparable per se.

- I feel Figure 2A shows too many different colors and thus becomes a bit difficult to read. Maybe this plot can be simplified. Not sure it is worthwhile showing the #branches explicitly here. This could be done in a supplemental figure.

We have removed the colors of the dots, as they indeed overlap with the colors related to the methods. We also changed the visual aspects of this figure to improve readability.

- The rationale part of the methods section is partially redundant with the introduction of the manuscript. This could be further Streamlined.

We have changed both the introduction and rationale section to decrease the redundancy.

- Page 4: "We leveraged here both the trees inferred by PHLOWER" - ...and STREAM - this sentence is incomplete.

- Page 5: "are differential expressed" - are differentially expressed.

- Page 16: Typo in pseudo-time

These sentences have been corrected.

- Please add software and package versions used.

We have added python dependencies information at the package description in github: <https://github.com/CostaLab/phlower>.

- The links in the data availability section were broken.

We apologize for this problem. We provided a temporary access link, which was made unavailable once we released the data a few days after the submission. The correct link is <https://doi.org/10.5281/zenodo.13860460>.

- Adding a google colab for the tutorial would make it more accessible.

Remarks on code availability:

I have had a look at the code but did not run it. I think the authors should make an example available through google colab to make it easier to explore.

We added now examples related to pancreas progenitor, neurogenesis as a tutorial. We organized the website also to make the location of tutorials on MEF, kidney organoids and simulated data examples more clear (<https://phlower.readthedocs.io/en/latest/tutorial.html>). We also add google collab notebooks for these. They can be found here:

These are also now available as notebooks in

MEF fibroblast to neuron:

<https://colab.research.google.com/drive/1Ck4KL3nIBHQTM-4SHtAiNnr2PFGvf80q?usp=sharing>

Simulated DLA10 tree:

<https://colab.research.google.com/drive/1UfOWIHNSGgiEbUhRU15ddT7rbXx6HO0r?usp=sharing>

Paper kidney:

<https://colab.research.google.com/drive/1izHA7aAn1aQrQ3caDEBPhgTRWN0rXPHC?usp=sharing>

Pancreas:

<https://colab.research.google.com/drive/1l-Lml94TXpVpQJSJaSwAZfslkqBumPOf?usp=sharing>

Neurogenesis:

<https://colab.research.google.com/drive/1rZByWRpJcPcaTJpH4lnUdgqDirZjY7us?usp=sharing>

Comment to referees

Reviewer #1:

Remarks to the Author:

All my questions have been well addressed. I have no further comments!

We thank the reviewer for his constructive review and comments.

Reviewer #2:

To address points 1 and 2 from the review, we have: 1) expanded the number of real scRNA-seq data sets; 2) adopted an additional evaluation measure based on aggregating the four previously used measures; 3) used Violin plots to show metric distributions; and 4) performed stratified analysis regarding the type of tree structure. All these approaches are also adopted in the Dynverse manuscript. This makes our results comparable and allows us to address the points raised by the referee.

Please also note that we have reorganized the supplementary figures (in 10 Extended Data Figure and 10 supplementary figures) as proposed by the Nature Methods editorial team. This included the merging of smaller figures, while keeping all panels of the previous manuscript. In the reply below, we refer to the new names of the Extended data and Supplementary figures.

1. There is confusion by the statements in the blue text. It was stated that PHLOWER is the top performer in all 8 combinations (two groups of datasets and four metrics). It does not seem to outperform things on Fig. 2F and H. I believe these metrics on real datasets are more important than those on simulated data.

PHLOWER has the highest average statistics in all evaluated measures/scenarios for simulated data sets, as well as for the previous and novel results on the real data sets. In both the current and the previous version of the manuscript, the average criteria was used to rank the boxplots (see legend of former and current Fig. 2). The reviewer refers to the fact that in our previous submission the median of competing approaches is higher than PHLOWER (old Fig. 2F, pCreode and old Fig. 2H for Slingshot). To address this, we changed the plots for figure 2 to violin plots, which was also adopted in Dynverse, as these better reflect the complex distributions of statistics. You can find below both plots when considering the simulated data (Reply Fig. 1), the previous analysis of real data (Reply Fig. 2); and the novel analysis of real data (Reply Fig. 3). We adopt violin plots in the novel version of Figure 2.

Reply Fig. 1 - Comparison of Violin plots vs. Boxplot graphs for simulated results.

Reply Fig. 2 - Comparison of Violin plots vs. Boxplot graphs for previous selection of real data sets.

Reply Fig. 3 - Comparison of Violin plots vs. Boxplot graphs for novel real data sets results (Figure 2B).

2. In the original Dynverse paper, PAGA and Slingshot significantly outperform all other methods (<https://www.nature.com/articles/s41587-019-0071-9/figures/2>). But Slingshot is no longer a high performer in this paper. Maybe this is because this paper uses a subset of read data in Dynverse with more complex structure. Related to this, why not also include the performance of PHLOWER on the full Dynverse benchmark and use the evaluation values of other methods in the evaluation table in Dynverse paper? Are the authors deliberately hiding something?

To clarify the selection of real scRNA-seq, here is a more detailed description. Dynverse provides altogether 110 real data sets (see https://static-content.springer.com/esm/art%3A10.1038%2F541587-019-0071-9/MediaObjects/41587_2019_71_MOESM4_ESM.xlsx). Of these 110 data sets, 55 correspond to single rooted structures (linear, trees, bifurcation and multifurcation). These 55 data sets were listed in the Supplementary Table 1. PHLOWER and competing methods are only able to infer single rooted trees; and Dynverse itself only uses selections of data sets in their benchmark (see Fig. 2 of Dynverse <https://www.nature.com/articles/s41587-019-0071-9>). We removed from our evaluation over simplistic datasets based on trees with only 1 or 2 branches. In the previous version of the manuscript, we further removed 13 data sets, which were based on sub-setting of cells from the same original planaria data set. This filtering was motivated due to bias of evaluating methods in sub-sampling of the same data. This selection was indicated in the previous methods section “4.7.2 Real scRNA-seq data sets” and the 55 data sets including filtering criteria were indicated in Supplementary Table 1.

In the current version of the manuscript, we have updated our benchmark to include all versions of the planaria data set. We also provide an “accuracy” score, which was used by Dynverse to

provide the final ranking of methods (see Fig. 2 of Dynverse manuscript). The novel results can be seen in the novel Fig. 2B and the Reply Fig 3. As you can appreciate, PHLOWER has higher average values in all evaluated data sets. Regarding the rankings, it is the top ranked approach in all evaluations with the exception of the 'location of cells', where monocle3 is ranked first and PHLOWER second (Extended Data Figure 2A).

Regarding the performance of Slingshot, we have stratified the analysis by type of tree structure as provided in Dynverse. These results can be seen in Reply Fig. 4. We observe that some methods have a tendency to relatively perform better in simpler structures (Slingshot), while PAGA performs relatively better in more complex structures. PHLOWER performs equally well on all evaluated structures (Reply Fig. 5A). Another interesting analysis, also performed in Dynverse, is to relate the size of the estimated trees vs. the reference tree. We observe that some methods tend to over/underestimate the tree complexity. Of note, this simple statistic does not indicate topological similarity between the estimated and reference tree, which is measured by the "Structure Similarity" index. Our results shown that Slingshot particularly underestimated the number of branches in the tree (Reply Fig. 5B), and that PHLOWER has best results for complex multi branched and large trees.

Reply Fig. 4 - A) We show the distribution of performance scores for real scRNA-seq data sets stratified by type of tree structure (equivalent to new Fig. 2A). B) Figure S4. Real data benchmarking stratified by tree structure. A) HIM, correlation, F1 branches. B) Distributions of topology size differences between predicted and reference structures. Positive values indicate that the predicted topologies are more complex than the references, while negative values indicate simpler trees were estimated. Best approaches should have a lowest coefficient of variation (mean/std). Methods are ranked by decreasing CV values.

Also, the fact that Slingshot was a better performer in simpler structures was already discussed in Dynverse paper. Moreover, the fact some approaches tend to estimate over simplistic trees was already well indicated in the analysis of the pancreas progenitors and neurogenesis previously requested by this referee. As can be observed in Supp. Fig. 2 and 3, Slingshot and most of the competing methods only predict a simple linear structure covering the main cell differentiation path; while methods performing well in our benchmarking (PAGA and Monocle3) perform a better job in finding smaller sub-branchings.

We hope that this clarifies the fact that PHLOWER is the best performing approach in the data sets evaluated in our manuscript, which focused on complex and multi branched trees. These better represent the challenges of complex differentiation systems as organoids or multicellular organisms. These results are now implemented in Fig. 2 and Extended Data Fig. 2B-C. We have also updated the results and expanded our discussion with some of the points raised above.

3. Although improved, many mathematical statements are still not accurate.

We are happy to overview/revise any remaining point regarding mathematical statements.

Minor things:

We went through the text and figures to fix typos and capitalization issues.

Reviewer #3:

Remarks to the Author:

The authors have done an admirable job at addressing all reviewer concerns, in particular as some reviewer comments were not very well supported.

We thank the reviewer for the constructive review process; and appreciation of our work.